# SiReRAG: Indexing Similar and Related Information for Multihop Reasoning

**Nan Zhang**♣† **Prafulla Kumar Choubey**◇ **Alexander Fabbri**◇ **Gabriel Bernadett-Shapiro**◇
**Rui Zhang**♣ **Prasenjit Mitra**♣ **Caiming Xiong**◇ **Chien-Sheng Wu**◇
♣The Pennsylvania State University   ◇Salesforce AI Research
{njz5124,rmz5227,pmitra}@psu.edu
{pchoubey,afabbri,gbernadettshapiro,cxiong,wu.jason}@salesforce.com

## ABSTRACT

Indexing is an important step towards strong performance in retrieval-augmented generation (RAG) systems. However, existing methods organize data based on either semantic similarity (similarity) or related information (relatedness), but do not cover both perspectives comprehensively. Our analysis reveals that modeling only one perspective results in insufficient knowledge synthesis, leading to suboptimal performance on complex tasks requiring multihop reasoning. In this paper, we propose SiReRAG, a novel RAG indexing approach that explicitly considers both similar and related information. On the similarity side, we follow existing work and explore some variances to construct a similarity tree based on recursive summarization. On the relatedness side, SiReRAG extracts propositions and entities from texts, groups propositions via shared entities, and generates recursive summaries to construct a relatedness tree. We index and flatten both similarity and relatedness trees into a unified retrieval pool. Our experiments demonstrate that SiReRAG consistently outperforms state-of-the-art indexing methods on three multihop datasets (MuSiQue, 2WikiMultiHopQA, and HotpotQA), with an average 1.9% improvement in F1 scores. As a reasonably efficient solution, SiReRAG enhances existing reranking methods significantly, with up to 7.8% improvement in average F1 scores. Our code is available at https://github.com/SalesforceAIResearch/SiReRAG.

## 1 INTRODUCTION

Retrieval-augmented generation (RAG) has shown strong potential in augmenting large language models (LLMs) with highly specialized and constantly updated knowledge (Lewis et al., 2020; Gao et al., 2024). Getting rid of fine-tuning LLMs, it is an efficient method for handling users' queries that require domain knowledge. A typical RAG pipeline may involve chunking, embedding, indexing, retrieval with queries, reranking, and LLM response generation (Wang et al., 2024).

The indexing step is a prerequisite. It focuses on organizing a large amount of data and serves as an upstream step of retrieval. For example, RAPTOR (Sarthi et al., 2024) shows a significant performance improvement by adding recursive summaries to text chunks of a dataset, which demonstrates the potential of adding synthesized information for retrieval. The added recursive summaries combine semantically similar information within a dataset. GraphRAG (Edge et al., 2024), on the other hand, indices an entire corpus via an entity-guided knowledge graph. It then constructs summaries from closely-related entities and their mentions, synthesizing the connections and relatedness among different pieces of information.

However, none of the existing methods address the importance of indexing from both similarity and relatedness sides, which limits a holistic understanding of the provided dataset. We define **similarity** as the semantic distance of text pieces and **relatedness** as the degree of connection of texts based on signals such as entities and propositions. Indexing similar and related information facilitates more comprehensive knowledge integration than indexing individual kind of information. As shown in

---

†Work done while interning at Salesforce AI Research.

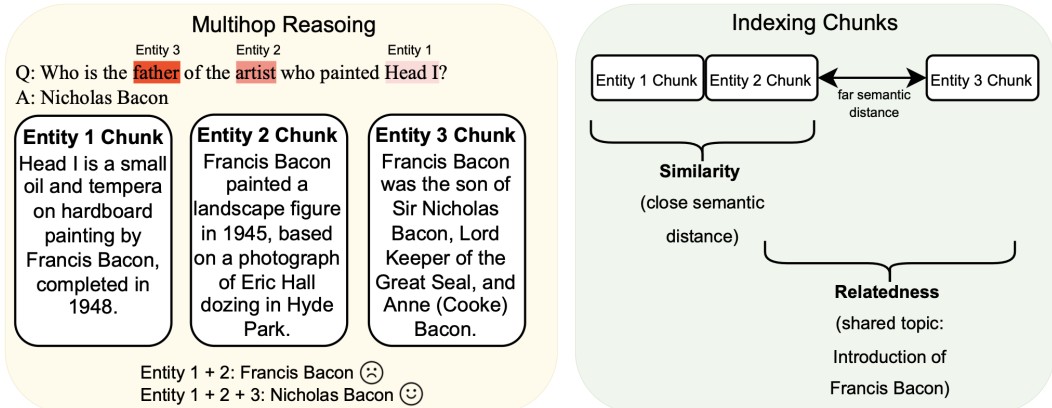

Figure 1: **Challenges of existing RAG indexing methods for multihop reasoning.** Entity 1 and 2 chunks contain similar information while entity 2 and 3 chunks contain related contents. Since synthesizing information only based on entity 1 and 2 (or entity 2 and 3) will lead to a higher probability of a wrong answer, an indexing method that considers both similarity and relatedness is needed to maximize retrieving relevant knowledge for multihop questions.

Figure 1, a complex question that involves two hops of reasoning requires the retrieval and synthesis of relevant entity chunks. For example, synthesizing entity 1 and 2 chunks would encourage LLMs to generate "Francis Bacon", which is a common mistake. Entity 1 and 2 chunks have close semantic distance. On the other hand, synthesizing entity 2 and 3 chunks would not maximize the probability of the correct answer even when entity 1 chunk is retrieved, since LLMs may struggle to reason about the painter of "Head I" in long-context environment (Liu et al., 2024). In this case, entity 2 and 3 chunks are related due to a shared topic. Therefore, it is important to synthesize both similar and related information in order to maximize the chance of retrieving relevant knowledge, in particular, for multihop reasoning questions. We verify the bottleneck of solely modeling similarity or relatedness through quantitative methods in Section 3 and demonstrate that neither perspective yields the optimal performance.

In this paper, we propose SIRERAG, which stands for RAG indexing of **si**milarity and **re**latedness as shown in Figure 2. On the similarity side, SIRERAG follows RAPTOR (Sarthi et al., 2024) to build a recursive tree based on chunk similarity. We adopt a shallow tree with 4 levels in total. On the relatedness side, SIRERAG first extracts entities (*e.g.*, "Sonnet 110" and "William Shakespeare") and fine-grained propositions (*e.g.*, "Sonnet 110 is one of 154 sonnets written by William Shakespeare.") from each text chunk/document using LLMs. We group these propositions into aggregated ones via entities, simply concatenating them with the original order in chunk/document. These proposition aggregates contain related information, because they mention shared entities. Then, recursive summaries with soft clustering are built on top of those aggregated propositions. Finally, we index both trees by flattening nodes in each tree for retrieval.

We show that SIRERAG is effective on a variety of multihop question answering (QA) datasets including MuSiQue (Trivedi et al., 2022), 2WikiMultiHopQA (Ho et al., 2020), and HotpotQA (Yang et al., 2018). We find SIRERAG consistently outperforms the strongest RAG indexing methods, achieving an average F1 improvement of at least 1.9%. We conduct an ablation study on several components in SIRERAG and observe that adding proposition aggregates to the similar information within a dataset yields the most improvement, which echoes our motivation. SIRERAG is also a reasonably efficient model, as it does not introduce many lengthy or redundant retrieval candidates.

## 2 RELATED WORK

**RAG** RAG is a framework that integrates retrieval mechanisms into generative models to enhance text generation by leveraging external knowledge. This concept has evolved from earlier retrieval-based methods such as DrQA (Chen et al., 2017) and DPR (Karpukhin et al., 2020). Instead of separating retrieval and generation phases, researchers also showed the potential of tightly coupling retrieval and generation into an end-to-end framework (Lewis et al., 2020).

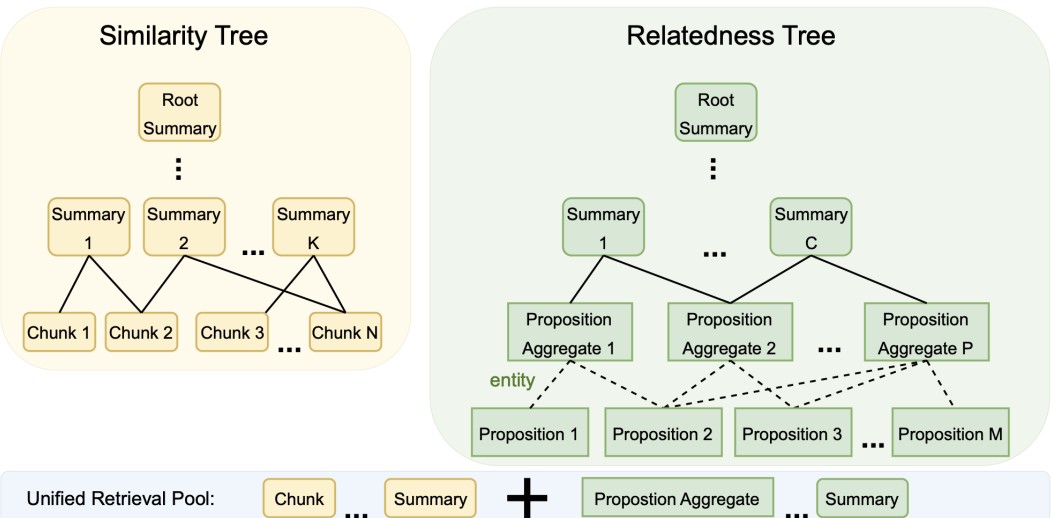

Figure 2: **SɪReRAG Tree.** We adopt RAPTOR (Sarthi et al., 2024) to construct the similarity tree (left). On the right, we construct the relatedness tree by clustering the propositions based on their entities to get proposition aggregates and having recursive summaries on top. Note that propositions are not included in the relatedness tree, so their connections to proposition aggregates are marked with dashed lines.

Recent advances in retrieval mechanisms include leveraging LLMs as retrievers (Yu et al., 2023; Sun et al., 2023) and exploring retrieval granularity such as proposition (Chen et al., 2023). Here each proposition is an atomic expression that contains a factoid presented in natural language, which is similar to the propositions used by SɪReRAG. Inspired by the idea of retrieval granularity, we combine several different text granularities (entity, propositions, text chunk, and summary) to index data. As a representative work on text segmentation, researchers proposed the frst supervised approach to generate hierarchical segmentation structures (Nair et al., 2023).

**RAG Indexing**   An earlier work (Lewis et al., 2020) shows the benefits of document index on the overall retrieval performance. Most recent works on RAG indexing include RAPTOR that builds a tree with recursive summaries (Sarthi et al., 2024), HippoRAG that leverages the hippocampal indexing theory of human long-term memory for deep knowledge integration (Gutiérrez et al., 2024), and GraphRAG that constructs an entity-guided knowledge graph (Edge et al., 2024). However, all these works overlooked the importance of considering both similarity and relatedness during indexing. Specifically, RAPTOR integrated knowledge only based on similarity, and the other two only considered relatedness to synthesize information. Although these three approaches achieved competitive performance on different kinds of datasets and HippoRAG has the same goal as ours (multihop reasoning benchmarks), SɪReRAG is fundamentally different from them in terms of the explicit incorporation of both similar and related knowledge.

## 3   BOTTLENECK OF SOLELY MODELING SIMILARITY OR RELATEDNESS

To verify our hypothesis of insufficient knowledge integration when solely modeling similarity or relatedness, we perform different kinds of clustering philosophies on a retrieval corpus of MuSiQue (Trivedi et al., 2022). Using the same retrieval corpus as HippoRAG (Gutiérrez et al., 2024), we obtain 1000 questions from the validation set of MuSiQue along with their candidate passage clusters (each cluster includes supporting and distractor passages). We include distractor passages for a more realistic setting, since they are semantically close and/or related to the supporting candidates. Treating these clusters as the gold labels for different queries, we run our own clustering on all passages based on either similarity or relatedness.

Following RAPTOR (Sarthi et al., 2024), we use Gaussian Mixture Models (GMMs) to perform soft clustering, assuming that a candidate passage can belong to multiple clusters. For similarity, we run GMMs on the deep representations of all passages to find semantically similar groups. For related-

Table 1: Coverage percentage between different clusters. MuSiQue clusters include supporting and distractor passages. The "all" setting treats both as gold, while "supporting only" uses only supporting passages as gold. We first show the coverage of supporting or all passages under two clustering philosophies. We then report the overlapping ratio between "supporting only" similarity and relatedness to motivate our work of combining both philosophies.

| | Similarity Coverage | | Relatedness Coverage | | Overlapping Ratio | |
|---|---|---|---|---|---|---|
| | Supporting Only | All | Supporting Only | All | Overlap@Similarity | Overlap@Relatedness |
| Coverage | 19.14% | 10.70% | 13.94% | 8.51% | 50.15% | 68.85% |

ness, we first extract the topic of each passage using OpenAI GPT-4o and then cluster passages based on the representations of their topics, because we assume that related passages share similar topics. To obtain deep representations of either passages or topics, we use OpenAI text-embedding-3-small for a balance of performance and efficiency. To evaluate the overlapping ratio among clusters, we convert every cluster into pairwise connections. For instance, given two clusters [1, 2, 3] and [3, 5], the resulting pairwise connections are as follows: "1-2", "1-3", "2-3", and "3-5". By computing the number of shared pairwise connections between gold clusters and predicted clusters, we aim to see how many pairwise connections from gold labels are covered by our two clustering philosophies, and we believe this coverage is a key indicator of knowledge integration for RAG indexing. We also report the overlapping ratio between "supporting only" similarity and relatedness to make a point of combining both clustering philosophies.

Table 1 shows the coverage of supporting only or all passages as gold labels. Higher coverage is obtained when we only use supporting passages as gold, which indicates the commonality of supporting ones with respect to distractors. Both philosophies are able to capture this commonality. However, all these percentage scores are low, with the highest one being around one-fifth of the total coverage (19.14%). This indicates a significant insufficiency in knowledge synthesis when we model similarity or relatedness solely. As a result, the chance of retrieving relevant knowledge for multihop reasoning questions could be suboptimal.

Taking a further step in the supporting-only setting, we find that only 50.15% of the correct similarity connections overlap with the correct relatedness connections. Correct similarity connections are those gold connections covered by similarity, while correct relatedness connections are those covered by relatedness. Conversely, 68.85% of the correct relatedness connections overlap with the correct similarity connections. In other words, predictions based on similarity or relatedness are not identical, and we can potentially leverage both to improve retrieval performance and facilitate a more comprehensive knowledge integration process. Although more customized clustering algorithms of each clustering philosophy can be proposed, combining both similarity and relatedness offers an effective and straightforward solution.

## 4 METHODOLOGY

We propose SIRERAG, a RAG indexing framework guided by similarity and relatedness. As shown in Figure 2, its left tree integrates information based on similarity while its right tree integrates information based on relatedness. As a first step, we study an alternative tree design to determine whether we can develop a generalized tree structure for similarity and relatedness trees, and beyond (Section 4.1). After the construction of the similarity tree, we extract propositions and their entities from our multihop reasoning dataset and perform clustering based on entities to synthesize related information (Section 4.2). Indexing separate similarity and relatedness trees (Section 4.3), SIRERAG explicitly models both kinds of information within our dataset.

### 4.1 EXPLORING A HIERARCHICAL STRUCTURE OF TREES

For efficiency, we stick to a tree structure to organize texts and explore whether a more structured tree design would offer performance improvement. RAPTOR placed all text chunks at the bottom level and recursive summaries at upper levels, but this design does not closely follow the commonsense

Table 2: QA performance of having hierarchical text chunks on the validation set of QuALITY. Due to randomization of clustering and generation temperature of LLMs, we run each indexing method for 5 times and compute their average and standard deviation of accuracies.

|  | Average Accuracy | Standard Deviation |
| --- | --- | --- |
| RAPTOR | 78.88 | 0.005 |
| Hierarchical Text Chunks | 78.76 | 0.004 |

definition of a tree (Zhang et al., 2002), where multiple levels of data abstraction are provided. The reason is that different text chunks of a document may showcase different levels of abstraction, which serves as the assumption of many document discourse trees (Maekawa et al., 2024; Liu et al., 2021). For example, text chunks of the introduction section of a research paper are more likely to be more abstractive than those from the methodology section. Therefore, we first explore placing different text chunks on different levels of a tree based on their levels of abstractiveness.

For this analysis, we choose QuALITY dataset (Pang et al., 2022) instead of multihop reasoning ones, because the documents of QuALITY are longer, resulting in more text chunks than using the other datasets. We prompt GPT-4o to identify a two-level hierarchy for all candidate chunks: low (text chunks describing fine-grained details about a topic) and high (text chunks giving an overview of a topic and summarizing fine-grained details) abstractiveness. Summary nodes will still start from the second level (the level above the bottom level), so the second level will contain both summary nodes (of the low-abstractive chunks) and high-abstractive chunks. As in RAPTOR, nodes of the third level and above are recursive summaries of their children. Using the same text chunks across two methods, we compare RAPTOR against our hierarchical design. Since we simply retrieve the top 10 relevant nodes via an embedding model (`text-embedding-3-small`) when a query arrives, the idea of hierarchical structure of chunks would affect summary nodes due to the differences of their children.

As shown in Table 2, we do not observe a clear trend of improvement from using hierarchical text chunks, even averaged over 5 different runs. The RAPTOR tree is relatively more efficient, as identifying hierarchical text chunks and document discourse trees would require additional computation. Thus, we conclude that a more structured tree design would not significantly improve performance, as long as the correct information is indexed to answer a question. Since RAPTOR uses Gaussian Mixture Models and representations of text chunks to perform clustering, it sets a nice example of integrating knowledge based on similarity. We adopt it to construct our similarity tree as shown in the left part of Figure 2.

## 4.2 SYNTHESIZING INFORMATION BASED ON RELATEDNESS

For relatedness, we need to synthesize information based on a different philosophy than RAPTOR. Because related information pieces always share some degree of connection (*e.g.*, overlapping subjects), we assume that two text pieces are related if they mention the same entity (e.g., person, location, product, etc). For example, entity 2 and 3 chunks in Figure 1 are unlikely to be clustered in the same group based on similarity, but since they both mention Francis Bacon, we are able to connect them together.

**Modeling Relatedness with Entity-Specific Propositions:** To effectively use entities for organizing related content, we first need to determine the appropriate granularity for text pieces. There are three main limitations with directly connecting entities to standard text chunks. First, a chunk often contains information beyond the scope of a specific entity, making it challenging to localize information about one entity, potentially adding noise. Second, aggregating all chunks in an indexing corpus for each entity can result in hundreds of thousands of tokens for each entity, which may lead to long context performance issues, such as losing critical information in the middle Liu et al. (2024) or experiencing low coverage and citation performance Laban et al. (2024). Third, linking with chunks will introduce redundancy as each chunk may be a part of multiple entity clusters. Therefore, inspired by recent works on retrieval granularity (Liu et al., 2023; Chen et al., 2023), we propose to use short entity-specific "propositions" to represent fine-grained knowledge about entities and build our relatedness tree.

Table 3: Key Statistics for extracted propositions and entities from MuSiQue, 2Wiki, and HotpotQA datasets. We show the number of chunks, propositions, entities, proposition aggregates, and the average, maximum, and minimum number of propositions per entity across the three datasets.

| Dataset | #Chunks | #Props. | #Ents | #Pops Aggs. | #Props per Entity | | |
|---|---|---|---|---|---|---|---|
| | | | | | Avg. | Max. | Min. |
| MuSiQue | 11,656 | 54,605 | 50,926 | 20,788 | 2.74 | 168 | 1 |
| 2Wiki | 6,119 | 27,697 | 29,490 | 11,108 | 2.49 | 195 | 1 |
| HotpotQA | 9,221 | 47,153 | 46,856 | 18,278 | 2.66 | 165 | 1 |

> *Proposition 1*: Drug sales are reaching record highs as new therapies are developed and approved.
> *Entities*: []
> *Proposition 2*: Drug sales for Eli Lilly's Mounjaro and Novo Nordisk's semaglutide are reaching record highs as new therapies are developed and approved.
> *Entities*: [Eli Lilly, Mounjaro, Novo Nordisk, Semaglutide]

Figure 3: Examples of propositions with and without associated entities.

**Extracting Propositions and Entities from Documents:** We define a proposition as "a factual statement describing important information (preferably about some entities) from a paragraph". We extract entities and propositions using the Distill-SynthKG pipeline (Choubey et al., 2024), adapting its SynthKG workflow. First, we rewrite chunks of 10K documents from the BAAI/IndustryCorpus[1] to resolve entity references, using `Meta-Llama-3-70B-Instruct`[2] (AI@Meta, 2024) with the rewriting prompt shown in Figure 6. Next, we prompt the same LLM to extract entities from these rewritten chunks (prompt is shown in Figure 7). After obtaining these entities, we again prompt the LLM to identify all relevant propositions and their associated entities (prompt is shown in Figure 8). We then consolidate the resulting propositions and entities to fine-tune `Mistral-7B-Instruct-v0.3`[3] (Jiang et al., 2023). This smaller fine-tuned model is subsequently used to extract propositions and their associated entities from our multihop datasets.

Our prompt for extracting propositions and entities does not require every proposition to have associated entities. When we prompt LLMs to generate entities for each proposition, they sometimes produce common nouns as entities for those propositions that lack actually associated entities. This can lead to the clustering of unrelated propositions based on common nouns, potentially introducing noise into the relatedness tree. For example, as illustrated in Figure 3, we prefer to avoid LLMs generating *Drug* as an entity for *proposition 1* due to its ambiguity. Subsequently, we exclude propositions without associated entities when constructing the relatedness tree, ensuring that only high-quality, entity-linked propositions are utilized. The Table 3 provides a detailed breakdown of key statistics for extracted propositions and entities from the MuSiQue, 2Wiki, and HotpotQA datasets.

**From Entity-specific propositions to Relatedness Tree:** We concatenate related propositions that share the same entity using exact match to form proposition aggregates. We ensure that all propositions from the same document are grouped together and maintain their original order. By treating these proposition aggregates as pseudo-documents, we apply the same clustering pipeline in RAPTOR to obtain recursive summaries at levels above them and build the relatedness tree. Given that most propositions involve multiple entities, each proposition is associated with several entity clusters, offering two key advantages. First, it effectively mimics soft clustering as a single proposition may belong to multiple aggregates. Secondly, when constructing recursive summaries within the RAPTOR framework, the shared propositions across different aggregates result in high embedding similarity, ensuring that these aggregates remain clustered together even at higher levels in the tree. This relatedness tree forms the right part of SIRERAG as shown in Figure 2. Note that we only index aggregated propositions instead of individual propositions for better inference efficiency.

---

[1] https://huggingface.co/datasets/BAAI/IndustryCorpus

[2] https://huggingface.co/meta-llama/Meta-Llama-3-70B-Instruct

[3] https://huggingface.co/mistralai/Mistral-7B-Instruct-v0.3

### 4.3 INDEXING SIMILARITY AND RELATEDNESS TREES

We propose to construct similarity and relatedness trees independently. This approach ensures that summary nodes in one tree do not access the clusters of the other, leading to a simpler design. There is another slightly complex design in which we allow summary nodes from one tree to access clusters from the other tree. This interaction may enable summary nodes in both trees to inform and enhance each other, improving their informativeness and consequently performance. However, this approach sacrifices the distinction between similarity and relatedness. Additionally, allowing cross-tree interaction leads to more nodes to cluster at each level as well as requires summarization based on a greater number of nodes per cluster, all of which increases the overall complexity of the system. We experimented with both settings and did not observe performance improvement as shown in Appendix A. Therefore, we opted for the simpler first implementation in our evaluation.

Flattening all tree nodes, we place them into a unified retrieval pool. In other words, regardless of a node's origin (*e.g.*, bottom or upper levels, similarity or relatedness trees), it is added to a single list containing all nodes.

## 5 EXPERIMENT SETUP

### 5.1 DATASETS

To demonstrate the effectiveness of SIRERAG, we select three representative multihop QA datasets: MuSiQue (Trivedi et al., 2022), 2WikiMultiHopQA (Ho et al., 2020), and HotpotQA (Yang et al., 2018). Using the same corpus as HippoRAG (Gutiérrez et al., 2024), we obtain 1000 questions from each validation set of these three datasets.

### 5.2 BASELINES

We select RAPTOR, HippoRAG, and GraphRAG as state-of-the-art retrieval baselines. As discussed above, RAPTOR integrates knowledge based on similarity while the other two approaches focus on relatedness. Specifically, HippoRAG has both indexing and retrieval components, and we use ColBERTv2 (Santhanam et al., 2022) as the retriever of HippoRAG due to its strongest QA performance reported. Although GraphRAG has a different goal (global questions directed at an entire dataset) than ours, we include it to show its performance on multihop QA datasets. Since the queries in our datasets ask fine-grained details, we use the local search function of GraphRAG instead of its global search. Additional details of GraphRAG are specified in Appendix F.

### 5.3 EVALUATION METRICS

We use exact match (EM) and F1 scores to measure the QA performance of different models. Both metrics evaluate how accurate a generated answer is with respect to the ground truth. Like RAPTOR (Sarthi et al., 2024), we do not assess retrieval performance directly. The reason is that both SIRERAG and RAPTOR create new candidates (*e.g.*, summary and proposition aggregate) in the retrieval pool, so it would be unfair to compare methods in terms of retrieval scores across different pools. Instead, QA performance is the best indicator of the overall capability of both RAG pipelines.

We use the average time per query (TPQ) and the time-pool efficiency ratio (TPER) to measure the efficiency of SIRERAG and RAPTOR, as both methods share a significant portion of their retrieval candidates. Average TPQ measures the average time (in seconds) taken to answer a query, and it represents the inference time of a method. For TPER, it computes the growth of total inference time with respect to the growth of the retrieval pool size between two methods:

$$\text{TPER} = \frac{\text{Inference-Time A}/\text{Inference-Time B}}{\text{Pool-Size A}/\text{Pool-Size B}} \tag{1}$$

Setting SIRERAG as method A and a baseline as method B, we aim to ensure that the growth of inference time does not scale proportionally with the increase in the retrieval pool size. The reason behind is that there are many efficiency considerations (*e.g.*, length and redundancy of retrieval candidates) beyond just the sheer number of retrieval candidates. Parallelization could also be designed

to retrieve candidates simultaneously from similarity and relatedness trees, thereby minimizing the effect of retrieval pool size. A TPER value less than 1 indicates reasonable efficiency, whereas a TPER value greater than 1 signifies low efficiency.

## 5.4 IMPLEMENTATION DETAILS

To generate final answer, we use GPT-4o and the same prompt (*"answer this question in as fewer number of words as possible."*) to answer queries for all methods, since we aim to control the instruction-following capabilities across all methods. We use either GPT-3.5-Turbo or GPT-4o as the choice of LLM if any methods involve LLM calls. We use OpenAI's `text-embedding-3-small` as the embedding model for all methods. During retrieval, we select top 20 candidates that match the provided query for all methods, because there is a large number of text chunks in our datasets and SIRERAG is expected to perform better when retrieving more due to the incorporation of proposition aggregates and their recursive summaries.

## 6 RESULTS AND ANALYSIS

Our results and analysis aim to answer the following research questions:

- **RQ 1**: How does SIRERAG compare against other state-of-the-art baselines (sec 6.1)?
- **RQ 2**: As an important contribution of SIRERAG, is considering both similarity and relatedness an effective method (sec 6.1 and 6.2)?
- **RQ 3**: What is the effect of each component in SIRERAG(sec 6.2)?
- **RQ 4**: What is the applicability of SIRERAG(sec 6.3)?
- **RQ 5**: With the addition of relatedness tree, is SIRERAG an efficient method (sec 6.4)?

Table 4: QA performance of SIRERAG and baselines. As elaborated in Section 5.4, GPT-4o is used to handle QA for all models, and we use two different LLMs (specified in the parentheses) to build indexing structures. We highlight the best scores using either LLM for indexing in green color.

| Model | MuSiQue | | 2Wiki | | HotpotQA | | Average | |
|---|---|---|---|---|---|---|---|---|
| | EM | F1 | EM | F1 | EM | F1 | EM | F1 |
| HippoRAG (GPT-3.5-Turbo) | 32.60 | 43.78 | 66.40 | 74.01 | 59.90 | 74.29 | 52.97 | 64.03 |
| RAPTOR (GPT-3.5-Turbo) | 35.30 | 47.47 | 54.90 | 61.20 | 58.10 | 72.48 | 49.43 | 60.38 |
| GraphRAG (GPT-4o) | 12.10 | 20.22 | 22.50 | 27.49 | 31.70 | 42.74 | 22.10 | 30.15 |
| RAPTOR (GPT-4o) | 36.40 | 49.09 | 53.80 | 61.45 | 58.00 | 73.08 | 49.40 | 61.21 |
| SIRERAG (GPT-3.5-Turbo) | 38.90 | 52.08 | 60.40 | 68.20 | 62.50 | 77.36 | 53.93 | 65.88 |
| SIRERAG (GPT-4o) | 40.50 | 53.08 | 59.60 | 67.94 | 61.70 | 76.48 | 53.93 | 65.83 |

## 6.1 OVERALL RESULTS

Our overall results are presented in Table 4. We show results on more datasets (single-hop QA, other multihop QA, and ambiguous questions) in Appendix E to show the generality of SIRERAG across various complex reasoning tasks. Besides quantitative scores, we also conduct our qualitative analysis in Appendix D.

**Improvement over baselines** SIRERAG delivers consistent improvement over RAPTOR, HippoRAG, and GraphRAG. With an exception on 2Wiki when comparing against HippoRAG, SIRERAG achieves significantly higher performance than indexing baselines (*e.g.*, approximately 5% higher than RAPTOR on average F1, up to 8.3% improvement of F1 on MuSiQue than HippoRAG, and more than 20% higher than GraphRAG on average EM and F1). This demonstrates the advantage of SIRERAG on multihop QA and modeling both similarity and relatedness. Specifically, SIRERAG outperforms RAPTOR due to the incorporation of a relatedness tree, and it has better overall performance than HippoRAG, because we explicitly model similarity while HippoRAG prioritizes relatedness signals such as nodes with the most edges. We see that HippoRAG is particularly strong on 2Wiki benchmark, which is also reported in its original paper. Thus, we believe 2Wiki is the best fit of HippoRAG, but it has lower performance scores than SIRERAG on other datasets.

Table 5: Ablation study of SIRERAG.

| Variants | MuSiQue | | 2Wiki | | HotpotQA | | Average | |
|---|---|---|---|---|---|---|---|---|
| | EM | F1 | EM | F1 | EM | F1 | EM | F1 |
| SIRERAG | 40.50 | 53.08 | 59.60 | 67.94 | 61.70 | 76.48 | 53.93 | 65.83 |
| (A): SIRERAG − Re. Summary | 37.50 | 50.38 | 57.70 | 65.75 | 61.20 | 75.99 | 52.13 | 64.04 |
| (B): SIRERAG + Proposition | 39.10 | 51.80 | 58.10 | 65.53 | 60.80 | 75.26 | 52.67 | 64.20 |
| (C): (B) − Aggregate | 34.70 | 47.82 | 53.20 | 60.22 | 58.90 | 73.38 | 48.93 | 60.47 |
| (D): (C) − Re. Summary | 33.90 | 46.19 | 53.00 | 59.13 | 57.00 | 71.09 | 47.97 | 58.80 |
| (E): Dual Clustering on Chunks | 34.80 | 47.32 | 53.50 | 59.93 | 56.60 | 71.84 | 48.30 | 59.70 |

One potential reason is that the entity-centric design of 2Wiki may be well-suited for HippoRAG, as noted in the HippoRAG paper.

As for GraphRAG, it considers relatedness solely, and it delivers the worst performance scores on our datasets. After a manual verification, we find that GraphRAG often provides "I don't know" answers, suggesting that it prefers not to give a concrete answer. Since GraphRAG is designed to handle query-focused summarization of an entire corpus, it is not the most competitive approach in terms of accuracy for existing multihop QA tasks.

**Effect of LLM choice** When comparing the performance of SIRERAG using GPT-4o as the LLM against itself using GPT-3.5-Turbo, we find the QA performance is not significantly affected. This phenomenon also holds on RAPTOR. Since the choice of LLM for SIRERAG and RAPTOR only affects summarization results, we believe GPT-3.5-Turbo is a sufficiently good option for both methods. This allows researchers to pursue a more cost-effective solution with SIRERAG for indexing.

## 6.2 ABLATION STUDY

To dissect SIRERAG, we perform a comprehensive ablation analysis as shown in Table 5. There are several variances, including (A) remove the recursive summary on the relatedness tree; (B) add all the propositions into the retrieval pool, and keep all aggregated propositions and recursive summary on the relatedness tree; (C) same as (B) but remove aggregated propositions; (D) same as (C) but further remove the recursive summary design on the relatedness tree.

**Entity clustering** In (E), we do not maintain a separate relatedness tree and add an additional clustering philosophy to the similarity tree. Specifically, each text chunk in the similarity tree is simplified to, 'This chunk mentions *entity 1* and *entity 2*,' if both entities are extracted by our LLM. We then run GMMs (the same clustering method as RAPTOR) on these simplified chunks. Once the clustering decisions are obtained, we group the original chunks as additional clusters and append these clusters to the similarity tree, allowing higher levels of the tree to incorporate both clustering philosophies. Since entities primarily determine the outcome of this additional clustering approach, we apply entity clustering to model relatedness on the similarity tree. This allows us to eliminate proposition aggregates in order to examine their utility.

**Findings** Overall, we observe performance drops across all variations, highlighting the effectiveness of our design for SIRERAG. First, the recursive summary on the relatedness tree proves beneficial, as seen in both (A) and (D). Interestingly, adding more propositions to retrieval negatively impacts performance, as shown in (B). This indicates adding redundant information into the retrieval pool hurts the QA performance, since we keep the aggregated propositions in SIRERAG. From (C), we also find that aggregated propositions are essential, with their removal resulting in a significant performance decline. This is an important indicator that adding grouped knowledge about relatedness to the similarity tree would offer improvements, which echoes the bottlenecks described in Section 3.

Both (A) and (B) have better performance than RAPTOR (GPT-4o) from Table 4, which indicates the advantage of proposition aggregates. In contrast, although (E) also models both similarity and relatedness, it exhibits a notable decline comparing against SIRERAG. This finding demonstrates the necessity of proposition aggregates of modeling relatedness. Because proposition aggregates reduce noise and information redundancy more effectively than text chunks as described in Section 4.2, they serve as an effective carrier of related dataset contents.

Table 6: Applicability of SIRERAG when a specific retrieval method is selected. We feed our non-indexing models with the retrieval pool of SIRERAG and see whether QA performance improves.

| Variants | MuSiQue | | 2Wiki | | HotpotQA | | Average | |
|---|---|---|---|---|---|---|---|---|
| | EM | F1 | EM | F1 | EM | F1 | EM | F1 |
| BM25 | 25.90 | 35.88 | 53.00 | 58.58 | 57.70 | 71.32 | 45.53 | 55.26 |
| RAPTOR + BM25 | 27.00 | 38.91 | 50.60 | 57.00 | 56.90 | 70.88 | 44.83 | 55.60 |
| SIRERAG + BM25 | 35.00 | 47.66 | 58.20 | 65.72 | 61.70 | 75.88 | 51.63 | 63.09 |
| ColBERTv2 | 34.00 | 46.80 | 52.90 | 59.48 | 59.00 | 73.42 | 48.63 | 59.90 |
| RAPTOR + ColBERTv2 | 34.20 | 47.17 | 50.30 | 57.51 | 57.90 | 72.31 | 47.47 | 59.00 |
| SIRERAG + ColBERTv2 | 38.10 | 51.32 | 56.70 | 64.74 | 60.90 | 75.72 | 51.90 | 63.93 |

Table 7: Efficiency of SIRERAG and RAPTOR. Since both methods share a significant portion of retrieval candidates, we designate SIRERAG as Method A and RAPTOR as Method B in the TPER columns, as defined in Equation 1.

| Model | MuSiQue | | 2Wiki | | HotpotQA | | Average | |
|---|---|---|---|---|---|---|---|---|
| | TPQ↓ | TPER↓ | TPQ↓ | TPER↓ | TPQ↓ | TPER↓ | TPQ↓ | TPER↓ |
| RAPTOR | 1.560 | - | 1.437 | - | 1.502 | - | 1.500 | - |
| SIRERAG | 2.653 | 0.600 | 1.974 | 0.499 | 2.319 | 0.517 | 2.315 | 0.539 |

## 6.3 APPLICABILITY OF SIRERAG

We analyze how applicable SIRERAG is when a specific retrieval method is chosen. Therefore, we select BM25 (Robertson & Walker, 1994) and ColBERTv2 (Santhanam et al., 2022) as additional reranking-based options. We run them on the retrieval pool of SIRERAG to demonstrate its utility. We show how SIRERAG can complement these non-indexing options in Table 6. Results show that having SIRERAG benefits both BM25 and ColBERTv2 significantly, which demonstrates the advantage of our solution as the upstream step of these methods. On the other hand, RAPTOR only improves the QA performance on MiSuQue while showing performance degradation on other datasets. Thus, the utility of SIRERAG surpasses that of RAPTOR in the context of multihop reasoning. We also apply SIRERAG on an iterative retrieval method called self-ask (Press et al., 2023) and obtain significant performance improvement as shown in Appendix B. Our method showcases wide applicability on multihop QA across various retrieval methods.

## 6.4 EFFICIENCY OF SIRERAG

As shown in Table 7, we compare the efficiency of SIRERAG and RAPTOR using the metrics described in Section 5.3. All the values listed involve the time taken to retrieve the top 20 candidates and prompt GPT-4o to answer the query.

RAPTOR requires less inference time than SIRERAG on average, which is expected due to SIRERAG's larger retrieval pool. However, with slightly longer inference time, SIRERAG has much better performance as discussed previously. To evaluate whether SIRERAG remains a reasonably efficient method, we compute its TPER values to measure its growth of total inference time relative to its growth of retrieval pool size. Since all its TPER values are well below 1, SIRERAG demonstrates reasonable efficiency without introducing many lengthy or redundant retrieval candidates.

## 7 CONCLUSION

In this paper, we identify the bottleneck of solely modeling similarity or relatedness when we need to index a multihop reasoning dataset for knowledge integration. To address it, we introduce SIRERAG, an innovative RAG indexing approach that considers both similarity and relatedness. SIRERAG delivers a consistent improvement over state-of-the-art indexing baselines across several multihop QA benchmarks.

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

## A    AN ALTERNATIVE DESIGN OF ALLOWING CROSS-TREE INTERACTION

We discuss an alternative design of combining similarity and relatedness trees. Specifically, this design combines nodes from both sides in the same pool for finding additional clusters and performing summarization at every tree level. In other words, we find additional clusters by concatenating the nodes of both trees, which considers cross-tree interaction instead of keeping them separate.

As shown in Table 8, the performance of considering cross-tree interaction is slightly lower than SIRERAG. Therefore, it is more efficient to keep trees separate in order to reduce the overall complexity of the system as discussed in Section 4.3.

## B    ADDITIONAL EXPERIMENT ON OTHER NON-INDEXING METHODS

Although there are many existing methods that work on multihop reasoning tasks, SIRERAG is about indexing corpus data under RAG setup. In other words, instead of being our baselines, other non-indexing works(Press et al., 2023; Islam et al., 2024) focus on other dimensions of improving performance on complex reasoning tasks.

Table 8: QA performance of two designs: separating similarity and relatedness trees (SIRERAG) and allowing cross-tree interaction.

| Variants | MuSiQue | | 2Wiki | | HotpotQA | | Average | |
|---|---|---|---|---|---|---|---|---|
| | EM | F1 | EM | F1 | EM | F1 | EM | F1 |
| SIRERAG | 40.50 | 53.08 | 59.60 | 67.94 | 61.70 | 76.48 | 53.93 | 65.83 |
| Cross-tree interaction | 40.20 | 53.06 | 58.30 | 65.25 | 60.30 | 75.71 | 52.93 | 64.67 |

Table 9: QA performance of closed-book and self-ask. We feed self-ask with the unified retrieval pool of SIRERAG and see whether performance benifits from that.

| Model | MuSiQue | | 2Wiki | | HotpotQA | | Average | |
|---|---|---|---|---|---|---|---|---|
| | EM | F1 | EM | F1 | EM | F1 | EM | F1 |
| Closed-book | 10.0 | 22.0 | 19.0 | 34.0 | 29.0 | 44.0 | 19.3 | 33.3 |
| Self-ask | 31.20 | 44.35 | 55.00 | 61.99 | 57.10 | 71.11 | 47.77 | 59.15 |
| SIRERAG + self-ask | 36.50 | 49.12 | 57.20 | 65.13 | 59.70 | 74.07 | 51.13 | 62.77 |

To further demonstrate the applicability of SIRERAG, we have run two additional sets of experiments: (1) the closed-book setting, and (2) an iterative retrieval method (Press et al., 2023) called self-ask specifically designed for multihop reasoning. The closed-book setting means to directly get the final answer from GPT-4o without any retrieval. For self-ask, we prompt GPT-4o in two iterations without using a search engine. In the first iteration, the model is prompted to propose follow-up questions and provide answers to them. In the second iteration, GPT-4o is instructed to answer the final question by incorporating the follow-up thought process. We feed the model with a one-shot example and 10 retrieved candidates that match the final question in both iterations. For the self-ask retrieval pool, we use either all text chunks or SIRERAG.

Similar to Table 6, our experiment in Table 9 on self-ask shows that SIRERAG can complement existing methods for optimal performance. We see that the closed-book setting yields the worst performance, which indicates that LLMs' parametric knowledge alone does not offer a decent performance on our datasets. Then, we see SIRERAG successfully improves the scores of self-ask, demonstrating its wide applicability. By leveraging SIRERAG's retrieval pool, we view our method as an augmentation to other non-indexing methods for multihop reasoning, rather than as a competitor.

Efficiency-wise, we show TPQ of using both SIRERAG and self-ask in Table 10. By having SIRERAG, the TPQ of self-ask increases by approximately 1.2 seconds over the three datasets. Since self-ask requires two iterations of LLM prompting in our implementation, the increase in TPQ is relatively small compared to the significant performance improvement brought by SIRERAG.

## C  RETRIEVAL POOL SIZE

The retrieval pool sizes of SIRERAG on MuSiQue, 2Wiki, and HotpotQA are 35070, 19100, and 29934 respectively. The retrieval pool sizes of RAPTOR on MuSiQue, 2Wiki, and HotpotQA are 12371, 6939, and 10031 respectively. SIRERAG's retrieval pool size is slightly less than three times the size of RAPTOR's. Considering the discussion on TPER in Section 6.4, we believe our method is reasonably efficient.

## D  AN EXAMPLE SIRERAG TREE

Using the question "who is the father of the artist who painted Head I?" as an example, we focus on the relevant part of the SIRERAG tree in Figure 4 to conduct our qualitative analysis.

For the multihop question in Figure 1 (correct answer: Nicholas Bacon), the MuSiQue corpus contains one relevant paragraph stating, "Francis Bacon was born on 22 January 1561 at York House

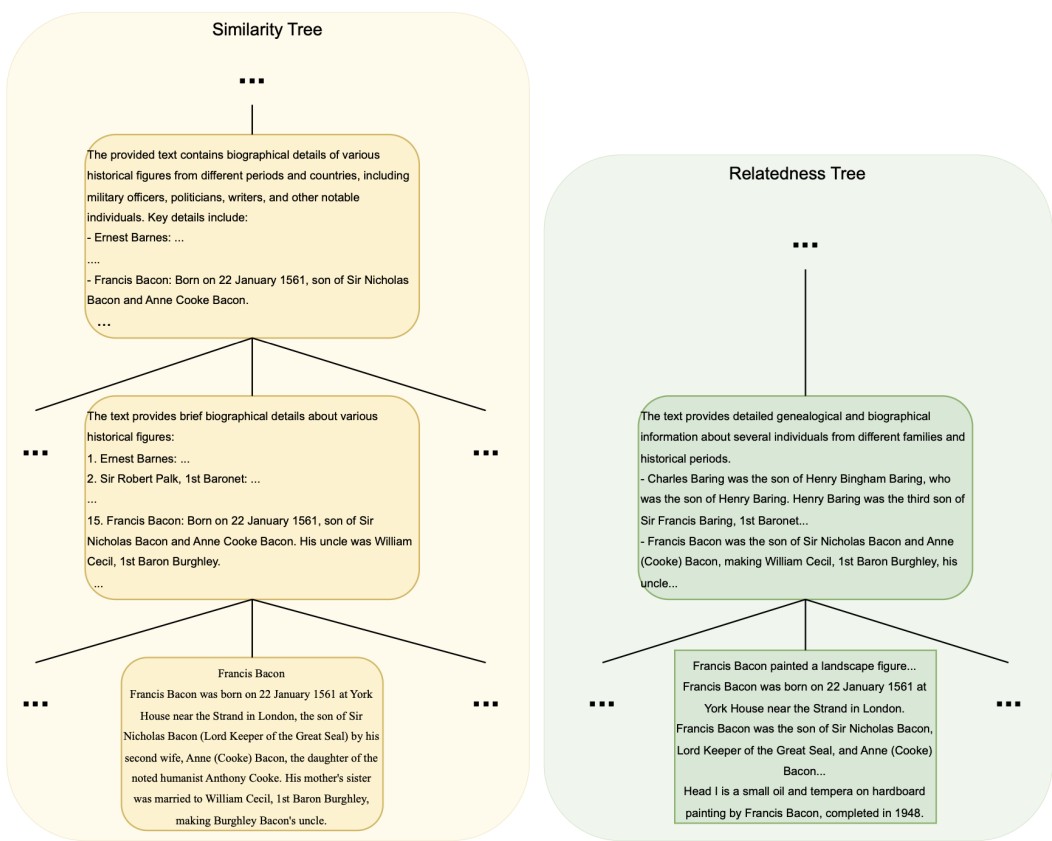

Figure 4: Relevant part of the SɪRERAG tree for the question: "who is the father of the artist who painted Head I?".

Table 10: Efficiency of using self-ask with SɪRERAG. We show TPQ on MuSiQue, 2Wiki, and HotpotQA datasets.

| Model | MuSiQue TPQ | 2Wiki TPQ | HotpotQA TPQ | Average TPQ |
|---|---|---|---|---|
| Self-ask | 2.72 | 2.21 | 2.29 | 2.41 |
| SɪRERAG + self-ask | 4.53 | 3.07 | 3.30 | 3.63 |

near the Strand in London, the son of Sir Nicholas Bacon...". This paragraph is a leaf node of the similarity tree shown in Figure 4. The similarity tree for the entire MuSiQue corpus has two more mentions (both mentions are in summary nodes) of Nicholas Bacon, one of which reads: "...Francis Bacon: Born on 22 January 1561, son of Sir Nicholas Bacon and Anne Cooke Bacon..." The addition of our relatedness tree adds two more mentions of Nicholas Bacon: one is in a proposition aggregate ("...Francis Bacon was the son of Sir Nicholas Bacon, Lord Keeper of the Great Seal, and Anne (Cooke) Bacon... Head I is a small oil and tempera on hardboard painting by Francis Bacon, completed in 1948..."), and the other one is a summary node ("The text provides detailed genealogical and biographical information about several individuals from different families and historical periods... Francis Bacon was the son of Sir Nicholas Bacon and Anne (Cooke) Bacon, making William Cecil, 1st Baron Burghley, his uncle..."). Thus, our similarity tree (RAPTOR tree) has three mentions of Nicholas Bacon, but none of them contains Head I information. SɪRERAG has five mentions of Nicholas Bacon, and one of them (the proposition aggregate) contains Head I information. This proposition aggregate groups several propositions together via the entity "Francis Bacon". Because this node is the only retrieval candidate that fully matches the question, retrieving it would maximize the chance of generating the correct answer. If we use the RAPTOR tree only, we will

not have this retrieval candidate. We believe this is an excellent example of how a comprehensive knowledge integration process can enhance the performance of RAG in multihop reasoning.

## E    PERFORMANCE ON SINGLE-HOP QA, MULTIHOP-RAG, AND AMBIGUOUS QUESTIONS

To showcase the generality of SIRERAG on more datasets of complex reasoning tasks, we run the comparison between SIRERAG and RAPTOR on single-hop questions and MultiHop-RAG dataset (Tang & Yang, 2024). We also try ASQA dataset (Stelmakh et al., 2022) that contains ambiguous factoid questions. For single-hop questions, we use MuSiQue dataset and collect all the decomposed questions of multihop queries. We filter out some decomposed questions if they are still multihop or are based on another question. As a result, we end up with 502 single-hop questions from MuSiQue. As for MultiHop-RAG, it is a more recent dataset. We filter all unanswerable questions and randomly select 350 "comparison" queries and 350 "inference" queries, which forms a pool of 700 queries in total.

Moreover, the primary difference between the multihop QA datasets and ASQA is that ASQA requires LLMs to reason across multiple perspectives (*e.g.*, disambiguated questions) of an ambiguous question and organize their generation into a coherent and detailed answer. We report scores on ASQA based on all 948 ambiguous questions of its development set.

Table 11 shows the performance scores of RAPTOR and our method. On the single-hop questions, SIRERAG still outperforms RAPTOR, but the lead narrows compared to the scores in Table 4. Since all queries in MuSiQue involve at least two hops, we observe that an increased number of reasoning hops positively impacts SIRERAG's performance. This is because single-hop questions may not require comprehensive knowledge synthesis, as they only involve retrieving the relevant chunks for the single hop. However, with more hops, we not only need to retrieve relevant chunks but also synthesize them comprehensively. SIRERAG also delivers better performance on MultiHop-RAG, which echoes our main experiment.

Table 11 also displays STR-EM (string exact match), Disambig-F1, and Disambiguation-Rouge metrics for ASQA dataset. Specifically, STR-EM and Disambig-F1 dissect RAG answers to ambiguous questions into multiple perspectives and measure their accuracy with respect to each disambiguated question. Disambiguation-Rouge serves as an overall statistic that incorporates both ROUGE (with respect to the references) and accuracy scores. We see SIRERAG delivers a consistent improvement over RAPTOR, which again demonstrates the benefit of adopting SIRERAG on ambiguous queries.

Our comprehensive selection of datasets demonstrates the generality and contribution of SIRERAG across various complex reasoning tasks, with the most significant improvement observed in multihop QA.

## F    ADDITIONAL GRAPHRAG DETAILS

We adhere to GraphRAG documentation[4] to construct its indexing structure and handle QA. Following the recommendation, we use its "auto tuning" to generate domain adapted prompts for the creation of its knowledge graph. We use the default values for most hyperparameters, except that we set the response type to "a few words" and do not include covariates during indexing.

## G    LLM PROMPTS

The prompt we use to perform summarization on a cluster of nodes is "summarize the provided text, including as many key details as needed". This prompt is the same as RAPTOR. In Section 3, we use 'identify the high-level topic of this paragraph as concise as possible" to extract the topic of each passage. As mentioned in Section 4.1, the prompt used for identifying a two-level hierarchy for all chunks is shown in Figure 5. As mentioned in Section 4.2, the LLM prompt for rewriting chunks is shown in Figure 6, the prompt for extracting named entities from rewritten chunks is shown in Figure 7, and the prompt for extracting propositions is shown in Figure 8.

---

[4] https://microsoft.github.io/graphrag/

Table 11: QA performance on single-hop questions, MultiHop-RAG, and ASQA.

| Model | Single-Hop | | MultiHop-RAG | | ASQA | | |
|---|---|---|---|---|---|---|---|
| | EM | F1 | EM | F1 | STR-EM | Disambig-F1 | Disambiguation-Rouge |
| RAPTOR | 73.90 | 80.67 | 85.43 | 86.37 | 51.47 | 40.30 | 42.72 |
| SIRERAG | 74.10 | 82.64 | 87.57 | 88.41 | 52.81 | 41.82 | 43.47 |

```
## Instructions
1. Group sentences based on their topic and abstractiveness.
2. For each sentence group define metadata consisting of topic and the abstractiveness level.
3. There are two abstractiveness levels:
    a. low: sentences describe fine-grained details about the topic
    b. high: sentences gives an overview of the topic, summarizing fine-grained details.
4. Ensure that each group consists of contiguous sentences.
5. Respond using the below json format.
{
  "group_1":
  {
    "sentences": [list of sentence ids],
    "topic": "topic of the group",
    "abstractiveness": "high"
  },
  "group_2":
  {
    "sentences": [list of sentence ids],
    "topic": "topic of the group",
    "abstractiveness": "low"
  },
  "group_3":
  {
    "sentences": [list of sentence ids],
    "topic": "topic of the group",
    "abstractiveness": "high"
  }
}
```

Figure 5: Prompt of identifying a two-level hierachy for all candidate chunks.

---

**Previous paragraph from Document**:
Gualala, the isolated Mendocino Coast town with a name that leaves most visitors tongue-tied, is on a new list of the 50 best places to live in the United States. Men's Journal magazine describes Gualala as an öutpost of adventure lifestyleïn its latest edition, which goes on sale today. The magazine describes Gualala (pronounced wa-LA-la by locals) as one of the below-the-radar places to a make a move on before the word gets out.There were five such cities. The others were Homer, Alaska; Newport, Vt.; Logan, Utah; and Walla Walla, Wash. Rolling Stone magazine's Jann Wenner publishes Men's Journal, which has a paid circulation of about 620,000. Gualala joined three other California communities on the magazine's list: Santa Cruz, Mammoth Lakes and Bishop. We were looking for places that combined affordability, proximity to outdoor adventure and a generally undiscovered quality of life,šaid Erica Kestenbaum, a spokeswoman for Men's Journal.

**Instruction**:
Rewrite the below paragraph by resolving all entity coreferences with the preceding paragraph from document.
- Resolve all inter-sentence pronoun references.
- Make sure that all pronouns in a sentence refers to some named entity with in the same sentence.
- Explicitly mention entity names wherever necessary to remove ambiguity from a sentence. Remember to make each sentence clear and unambiguous.
- For each entity, use only the one most informative name.
- Do not generate anything except the rewritten paragraph.

**Paragraph**:
She said isolation played a factor. Ïn Northern California, it's particularly difficult to find a beautiful coastal setting that isn't entirely overrun,šhe said. Gualala residents Monday were largely unaware of the magazine listing or the attention it could bring to the old logging town turned tourist center. A few coastal residents chuckled about any notion of affordability, given an influx of newcomers who've driven the median housing price to $580,000 compared to the median family income of $47,778. Others recalled an era when the Gualala region was better known for the logging of ancient redwoods, marijuana growing and boisterous beer drinking at the historic Gualala Hotel. Still there was a certain pride to the magazine's designation. Yvette White, a 25-year resident who works at the Gualala Sport; Tackle shop, said she's proud her town made it on the list.

**Output**:
Erica Kestenbaum said isolation played a factor. Ïn Northern California, it's particularly difficult to find a beautiful coastal setting that isn't entirely overrun,Ërica Kestenbaum said. Gualala residents Monday were largely unaware of the Men's Journal magazine listing or the attention it could bring to the old logging town turned tourist center. A few coastal residents of Gualala chuckled about any notion of affordability, given an influx of newcomers who've driven the Gualala's median housing price to $580,000 compared to the median family income of $47,778. Other Gualala residents recalled an era when the Gualala region was better known for the logging of ancient redwoods, marijuana growing and boisterous beer drinking at the historic Gualala Hotel. Still there was a certain pride to the Men's Journal magazine's designation. Yvette White, a 25-year Gualala resident who works at the Gualala Sport; Tackle shop, said she's proud her town made it on the list.

**Previous paragraph from Document**: [previous paragraph]

**Instruction**:
Rewrite the below paragraph by resolving all entity coreferences with the preceding paragraph from document.
- Resolve all inter-sentence pronoun references.
- Make sure that all pronouns in a sentence refers to some named entity with in the same sentence.
- Explicitly mention entity names wherever necessary to remove ambiguity from a sentence. Remember to make each sentence clear and unambiguous.
- For each entity, use only the one most informative name.
- Do not generate anything except the rewritten paragraph.

**Paragraph**: [paragraph ]
**Output**:

---

Figure 6: Prompt for rewriting a paragraph (*e.g.*, a document chunk) by resolving entity coreferences.

```
Extract all named entities from the document. Also generate the type for each entity.
Instructions
- Generate only the most informative name for each named entity. Example: if John P., Parker, John Parker are coreferential, only generate John Parker.
- Use your best understanding best on the domain of paragraph to decide appropriate entity types.
- Respond using json format provided below.

{
    "n1":{"name": "entity_name", "type": "entity_type_label"},
    "n2":{},
}

Below is an example for reference.
Paragraph: Tucked into Eli Lilly's year-end earnings report, the company revealed positive results from Synergy-NASH—its phase 2 study of tirzepatide in
adults in nonalcoholic steatohepatitis (NASH), also known as metabolic dysfunction-associated steatohepatitis (MASH).
Output:

{
    "n1": {"name": "Eli Lilly", "type": "Organization"},
    "n2": {"name": "Synergy-NASH", "type": "Clinical Trial"},
    "n4": {"name": "tirzepatide", "type": "Drug"},
    "n5": {"name": "nonalcoholic steatohepatitis", "type": "Disease"},
    "n6": {"name": "metabolic dysfunction-associated steatohepatitis", "type": "Disease"},
    "n7": {"name": "year-end earnings report", "type": "Document"}
}
```

Figure 7: Prompt for extracting entities from a document (*e.g.*, a rewritten chunk).

```
Extract all facts from the document. For each fact, also generate all semantic triplets.
Instructions
- Consistently use the most informative name for each named entity in all facts and triplets.
- Avoid pronouns or ambiguous references in facts and triplets. Instead, directly include all relevant named entities in facts.
- Ensure that each semantic triplet contains head entity, predicate, and tail entity.
- Ensure that at least one (preferably both) entity in each semantic triplet is present in the given entities list.
- Respond using json format provided below:

{
    "f1":{
        "fact": "A factual statement describing important information (preferably about some entities) from the paragraph",
        "triplets: [["entity 1", "predicate", "entity 2"], ["entity 1", "predicate", "entity 3"]]
    },
    "f2":{},
}

Below is an example for reference.
Paragraph: Locked in a heated battle with Novo Nordisk's semaglutide franchise, Eli Lilly's tirzepatide is beginning to come into its own—both with regards
to sales and amid attempts to show the dual GIP/GLP-1 agonist can strike out beyond diabetes and obesity. As Mounjaro, tirzepatide won its first FDA nod
in Type 2 diabetes back in May 2022. An obesity approval followed last November, with that formulation of tirzepatide adopting the commercial moniker
Zepbound. In 2023's fourth quarter, Mounjaro generated a whopping $2.2 billion in sales, a nearly eight-fold increase over the $279 million it pulled down
during the same stretch in 2022. Year-to-date, the drug brought home around $5.2 billion in revenues, Lilly said in an earnings release Tuesday. Zepbound,
for its part, generated $175.8 million during its first quarter on the market. Overall, Lilly reeled in around $9.4 billion in fourth-quarter sales, growing 28%
over the $7.3 billion it made for the quarter in 2022.
Entities: Eli Lilly, Novo Nordisk, Tirzepatide, Semaglutide, GLP-1, GIP, FDA, Mounjaro, Zepbound
Output:

{
    "f1": {
        "fact": "Eli Lilly's tirzepatide is competing with Novo Nordisk's semaglutide franchise.",
        "triplets": [["Eli Lilly", "competing with", "Novo Nordisk"], ["Tirzepatide", "is competing with", "Semaglutide"]]
    },
    "f2": {
      "fact": "Eli Lilly is trying to show tirzepatide, the dual GIP/GLP-1 agonist, can strike out beyond diabetes and obesity.",
        "triplets": [["Eli Lilly", "is trying to show", "Tirzepatide"], ["Tirzepatide", "is a", "dual GIP/GLP-1 agonist"],
                    ["Tirzepatide", "can treat beyond", "Diabetes"], ["Tirzepatide", "can treat beyond", "Obesity"]]
    },
    "f3": {
        "fact": "Tirzepatide, under the brand name Mounjaro, received its first FDA approval for Type 2 diabetes in May 2022.",
        "triplets": [["Tirzepatide", "branded as", "Mounjaro"], ["Mounjaro", "won", "FDA approval"],
                    ["FDA approval", "for",  "Type 2 diabetes"], ["FDA approval", "was in", "May 2022"]]
    },
    "f4": {
        "fact": "Tirzepatide, under the brand name Zepbound, received an obesity approval in November 2022.",
        "triplets": [["Tirzepatide", "was branded as", "Zepbound"], ["Zepbound", "received", "Obesity approval"],
                    ["Obesity approval", "was in", "November 2022"]]
    },
    "f5": {
      "fact": "Mounjaro generated $2.2 billion in sales in the fourth quarter of 2023, an eight-fold increase from the $279 million
        during the same period in 2022.",
        "triplets": [["Mounjaro", "2023's fourth quarter sales", "$2.2 billion sales"],
        ["Mounjaro", "2022's fourth quarter sales", "$279 million"]]
    },
    "f6": {
      "fact": "Mounjaro brought in around $5.2 billion in revenues year-to-date in 2023, Lilly said in an earnings release Tuesday",
        "triplets": [["Mounjaro", "2023 sales year-to-date", "$5.2 billion revenues"]]
    },
    "f7": {
        "fact": "Zepbound generated $175.8 million in sales in its first quarter on the market.",
        "triplets": [["Zepbound", "first quarter sales", "$175.8 million"]]
    },
    "f8": {
      "fact": "Eli Lilly's fourth-quarter sales were around $9.4 billion, a 28% increase over the $7.3 billion during the same
        period in 2022.",
        "triplets": [["Eli Lilly", "2023 fourth-quarter sales", "$9.4 billion,"],
        ["Eli Lilly", "2022 fourth-quarter sales", "$7.3 billion,"]]
    }
}
```

Figure 8: Prompt for extracting propositions (*e.g.*, facts) and their corresponding entities from a document.

