# OpenReview forum: "SiReRAG: Indexing Similar and Related Information for Multihop Reasoning"
_ICLR.cc/2025/Conference — ICLR 2025 Poster_

### Official Review · Reviewer_F1mV · 2024-10-28

**Soundness:** 2
**Presentation:** 2
**Contribution:** 2
**Rating:** 5
**Confidence:** 4

**Summary:**

Indexing is critical for IR or RAG systems. In this paper, the authors propose a RAG indexing method which considers both semantic similarity and relatedness for better retrieval, named SIRERAG. Specifically, for similarity-based indexing this paper constructs a similarity tree via recursive summarization, and for relatedness this paper constructs a relatedness tree via entity and proposition extraction and grouping. Experimental results show some performance improvement on multi-hop QA datasets over previous similarity-based baselines and relatedness-based baselines.

**Strengths:**

In general, I think it is helpful to leverage multiple types of relevance for better retrieval results, and the experimental results also verify its effectiveness. It is also helpful to investigate and identify that similarity-only is not enough for complex QA tasks (although this is not a new discovery as many previous studies in traditional IR studies).

**Weaknesses:**

1. Because the similarity-based indexing component just follows RAPTOR (Sarthi et al., 2024), I think the main contribution of this paper maybe the relatedness-based indexing component.  Unfortunately, I found the relatedness-based indexing algorithm is ad hoc. Firstly, the relatedness are modeled using entity-specific propositions, which I think is over-specialized to multi-hop QA tasks(where the hop is just entity-entity associations and this is why the proposed method can achieve performance improvements). However, I think entity-specific propositions may not a good decision for many other complex reasoning tasks, and the authors should explain and verify the effectiveness and generality in more tasks. Secondly, I found there are many heuristic decisions, such as how to filter proportions, how to resolve entity references, etc.
2. The similarity and relatedness trees are constructed and used independently, which I think is  straightforward, it is important to consider the interaction between different relevance scores.
3. There are many multi-hop QA baselines, such as iterative RAG-based, agent-based, etc. The authors should compare them for more convincing experimental results.
4. The writings of this paper should be improved.

**Questions:**

1. The experimental results on other multi-hop reasoning tasks.
2. The performance using other relatedness-based trees (e.g., entity-based, or entity pair-based)

---

> ### Author Response · Authors · 2024-11-25
> **Response to Reviewer F1mV (Part 1)**
>
> We thank the reviewer for the helpful feedback! We plan to submit another draft that incorporates all reviewers’ comments before the discussion phase ends. Focusing on multihop reasoning, we are, to the best of our knowledge, the first to verify the importance of incorporating both similarity and relatedness signals and to implement this concept in RAG indexing.
>
> **Weakness 1**:
>
> > “Because the similarity-based indexing component just follows RAPTOR (Sarthi et al., 2024), I think the main contribution of this paper maybe the relatedness-based indexing component.”
>
> To the best of our knowledge, our work is the first indexing method of modeling both similarity and relatedness in RAG setup for a more comprehensive knowledge integration. Our baselines (RAPTOR, HippoRAG, and GraphRAG) consider only a single perspective. For example, HippoRAG and GraphRAG are relatedness-based, constructing noun- or entity-guided knowledge graphs. In contrast, we find that modeling both perspectives enables a more effective knowledge integration process, as detailed in Section 3 (adopting a clustering strategy based on a single perspective covers a remarkably low percentage of supporting passages). Our quantitative results in Section 6 echoes this motivation.
>
> Although we are motivated, it also requires nontrivial efforts to realize our high-level idea as a RAG indexing method. Firstly, we decide to focus on multihop reasoning as our scope, as knowledge integration or synthesis is vital to this kind of task as discussed in Figure 1. Our method does not show a significant improvement on other kinds of datasets that involve less reasoning such as QASPER. Then, we need to finalize an effective and efficient structure (e.g., tree/knowledge graph) to index corpus data. RAPTOR tree is a straightforward idea, but it deviates from the commonsense understanding of a tree structure [1]. As noted in line 207, while multiple levels of data abstraction can exist within a document, RAPTOR places all text chunks at the bottom level. This approach conflicts with many studies on document discourse trees [2]. Then, we explore the hierarchical structure of an indexing tree. However, we do not see a clear trend of improvement by doing so (Table 2). So, we use RAPTOR as the generalized tree structure for similar and related information. We believe our exploration and observation are valuable for future work.
>
> [1] Zhang, J., Silvescu, A., & Honavar, V. (2002). Ontology-driven induction of decision trees at multiple levels of abstraction. In Abstraction, Reformulation, and Approximation: 5th International Symposium, SARA 2002 Kananaskis, Alberta, Canada August 2–4, 2002 Proceedings 5 (pp. 316-323). Springer Berlin Heidelberg.
>
> [2] Maekawa, A., Hirao, T., Kamigaito, H., & Okumura, M. (2024, March). Can we obtain significant success in RST discourse parsing by using Large Language Models?. In Proceedings of the 18th Conference of the European Chapter of the Association for Computational Linguistics (Volume 1: Long Papers) (pp. 2803-2815).

---

> ### Author Response · Authors · 2024-11-25
> **Response to Reviewer F1mV (Part 2)**
>
> > “Unfortunately, I found the relatedness-based indexing algorithm is ad hoc.”
>
> As indicated above, we adhere to a rigorous thought process. In order to demonstrate a principled design of our method, we have our ablation study in Section 6.2. Here, we also add an additional "entity clustering” experiment to showcase the effectiveness of entity-specific propositions on multihop reasoning tasks.
>
> We perform clustering directly on text chunks by using entities (entity clustering). Specifically, we do not maintain a separate relatedness tree and add an additional clustering philosophy to the similarity tree. Each text chunk in the similarity tree is simplified to “This chunk mentions entity 1 and entity 2” if both entities are extracted by our LLM. We then run GMMs (the same clustering method as RAPTOR) on these simplified chunks. Once the clustering decisions are obtained, we group the original chunks as additional clusters and append these clusters to the similarity tree, allowing higher levels of the tree to incorporate both clustering philosophies. Since entities primarily determine the outcome of this additional clustering approach, we apply entity clustering to model relatedness on the similarity tree. This allows us to eliminate proposition aggregates in order to examine their utility. Performance is shown below.
>
> |                        | MuSiQue EM | MuSiQue F1 | 2Wiki EM | 2Wiki F1 | HotpotQA EM | HotpotQA F1 | Average EM | Average F1 |
> |------------------------|------------|------------|----------|----------|-------------|-------------|------------|------------|
> | **SiReRAG**           | **40.50**  | **53.08**  | **59.60**| **67.94**| **61.70**   | **76.48**   | **53.93**  | **65.83**  |
> | **Dual clustering on Chunks** | 34.80     | 47.32     | 53.50    | 59.93    | 56.60       | 71.84       | 48.30      | 59.70      |
>
> Although this dual clustering method also models both similarity and relatedness, it exhibits a notable decline in comparison to SiReRAG. This finding empirically demonstrates the necessity of proposition aggregates of modeling relatedness. Because proposition aggregates reduce noise and information redundancy more effectively than text chunks as described in Section 4.2, they serve as an effective carrier of related dataset contents. We will add this table in our revised draft that will be submitted later.
>
> > “Firstly, the relatedness are modeled using entity-specific propositions, which I think is over-specialized to multi-hop QA tasks(where the hop is just entity-entity associations and this is why the proposed method can achieve performance improvements). However, I think entity-specific propositions may not a good decision for many other complex reasoning tasks, and the authors should explain and verify the effectiveness and generality in more tasks.”
>
> First, Entity-Propositions can cover diverse type of entity-specific questions: e.g., factual or single-hop questions, multi-hop reasoning, information aggregation and summarization (e.g., which clubs in NYC offer boxing studio? summarize all important contributions of WHO, etc.). Datasets we used in our evaluation already include these different types of reasoning: e.g., comparison question, compositional question, inference question, temporal reasoning question, etc. For instance, MuSiQue [3] includes 5 reasoning structures [Table 1].
>
> Next, we want to highlight that entity-specific RAG approaches cover a significant proportion of RAG use cases, and is a growing research area, as shown by recent works like HippoRAG [1] and GraphRAG [2]. These methods exclusively explore entity graph-based RAG approaches, underscoring their growing relevance and utility in answering complex questions. Also, relatedness tree (based on entity propositions) is complementary to chunk-based RAPTOR. So, it should assist on complex questions requiring evidences/ propositions about entities, without impacting other non-entity specific questions.
>
> Furthermore, beyond our evaluation on three multi-hop QA datasets in paper, SiReRAG also outperforms RAPTOR on MultiHopRAG, as discussed in our response to Question 1 below. This further highlights that SiReRAG is generalizable to any complex multi-hop reasoning tasks.
>
> Lastly, propositions aggregation according to entity can be expanded to any other discourse units, e.g. Topic which could make an interesting future exploration.
>
> [1] HippoRAG https://arxiv.org/abs/2405.14831
>
> [2] GraphRAG https://arxiv.org/pdf/2404.16130
>
> [3] MuSiQue https://arxiv.org/pdf/2108.00573

---

> ### Author Response · Authors · 2024-11-25
> **Response to Reviewer F1mV (Part 3)**
>
> > “Secondly, I found there are many heuristic decisions, such as how to filter proportions, how to resolve entity references, etc.”
>
> With respect to heuristic decisions, it is necessary to incorporate them in order to maximize our performance. For example, we notice that an LLM could effectively resolve entity references by rewriting documents. Otherwise, without rewriting documents, related propositions may not be grouped together due to inconsistent names of some entities, which weakens our method. For example, in the MuSiQue corpus, there is a sentence about the “Head I” painting: “It is the first of Bacon's paintings to feature gold background railings or bars, which later became a prominent feature of his 1950s work, especially in the papal portraits where they often appeared as enclosures or cages around the figures.” Without rewriting “Bacon” to “Francis Bacon” prior to indexing, it would be difficult to identify who 'Bacon' refers to, since the example in Figure 1 includes two “Bacon”s: Francis Bacon and Nicholas Bacon. As a result, performance of generating the correct answer (Nicholas Bacon) would be lower than without rewriting documents.
>
> **Weakness 2**: Thanks for your suggestion. We discussed the motivation of separating similarity and relatedness trees and an alternative design of combining them in Section 4.3. Specifically, in order to concatenate similarity and relatedness trees, this alternative design combined nodes from both sides in the same pool for finding additional clusters and performing summarization at every tree level (line 317). In other words, we find additional clusters by concatenating the nodes of both trees, which considers cross-tree interaction instead of keeping them separate. The performance of doing so is shown below.
>
> |                                         | MuSiQue EM | MuSiQue F1 | 2Wiki EM | 2Wiki F1 | HotpotQA EM | HotpotQA F1 | Average EM | Average F1 |
> |-----------------------------------------|------------|------------|----------|----------|-------------|-------------|------------|------------|
> | **SiReRAG**                             | **40.50**  | **53.08**  | **59.60**| **67.94**| **61.70**   | **76.48**   | **53.93**  | **65.83**  |
> | **Concatenating Nodes of Similarity and Relatedness** | 40.20     | 53.06     | 58.30    | 65.25    | 60.30       | 75.71       | 52.93      | 64.67      |
>
> The performance of considering cross-tree interaction is slightly lower than SiReRAG. Therefore, it is more efficient to keep them separate in order to reduce the overall complexity of the system as discussed in Section 4.3. Since we have shown the benefit of indexing both similar and related information, we do not incorporate cross-tree interaction due to effectiveness and efficiency. We will add this table in our revised draft that will be submitted later.
>
> **Weakness 3**: Although there are many existing methods that work on multihop reasoning tasks, SiReRAG is about indexing corpus data under RAG setup. In other words, instead of being our baselines, other existing works focus on other dimensions of improving performance on complex reasoning tasks. To the best of our knowledge, we have selected a wide variety of RAG indexing baselines to demonstrate our claims.
>
> However, as suggested by Reviewer ymw3, we have run an additional set of experiments on an iterative retrieval method (Self-Ask) specifically designed for multihop reasoning. In addition to Table 6, these experiments show how SiReRAG can complement existing methods for optimal performance.
>
> |               | MuSiQue EM | MuSiQue F1 | 2Wiki EM | 2Wiki F1 | HotpotQA EM | Hotpot F1 | Average EM | Average F1 |
> |---------------|------------|------------|----------|----------|-------------|-----------|------------|------------|
> | Self-Ask      | 31.20      | 44.35      | 55.00    | 61.99    | 57.10       | 71.11     | 47.77      | 59.15      |
> | SiReRAG + Self-Ask | **36.50** | **49.12** | **57.20** | **65.13** | **59.70**  | **74.07** | **51.13**  | **62.77**  |
>
> Self-Ask uses novel inference questions to decompose a multihop query. By running Self-Ask on the retrieval pool of SiReRAG, we obtain performance improvement on all three datasets with an average improvement of approximately 3.6% in F1 score.
>
> **Weakness 4**: We will be more than happy to clarify any points! We are also revising our draft for better clarity and will submit the pdf shortly.

---

> ### Author Response · Authors · 2024-11-26
> **Questions Continued..**
>
> **Question 1**: Same as HippoRAG, we reported performance on MuSiQue, 2Wiki, and HotpotQA in our original submission. Following your suggestion, we are adding MultiHopRAG as another multihop reasoning task. We show the performance of SiReRAG and RAPTOR on this newer dataset. We filter all unanswerable questions and randomly select 350 “comparison” queries and 350 “inference” queries, which forms a pool of 700 queries in total.
>
> |         | MultiHopRAG EM | MultiHopRAG F1 |
> |---------|----------------|----------------|
> | RAPTOR  | 59.90          | 60.35          |
> | SiReRAG | **61.00**      | **61.69**      |
>
> We notice that MultiHopRAG is a relatively easier dataset than our selected ones in our paper, because both methods provide the correct answers in most cases. Notably, SiReRAG still delivers better performance, which echoes our main experiments. Our comprehensive selection of datasets demonstrates the generality of SiReRAG on multihop reasoning tasks.
>
> **Question 2**: Thanks a lot for your suggestion! We show the performance of entity clustering in our response to weakness 1, which demonstrates that an entity-based relatedness tree would offer suboptimal performance.
>
> We hope our response has addressed all your concerns and kindly hope you can consider increasing your scores.

---

> > ### Comment · Reviewer_F1mV · 2024-11-28
> >
> > Thanks for the rebuttal. Although the authors addressed some of my concerns, I will raise my score but my main concerns are still here that the relatedness indexing tree is helpful but specialized for specific tasks, therefore it is obvious to see some performance improvements in these tasks.

---

> > > ### Author Response · Authors · 2024-12-03
> > >
> > > Thanks a lot for your response! In our general post, we provided additional results on a different dataset called ASQA to demonstrate the generality of SiReRAG on complex reasoning tasks. Thanks again for your valuable feedback!

---

### Official Review · Reviewer_gpao · 2024-11-03

**Soundness:** 4
**Presentation:** 3
**Contribution:** 3
**Rating:** 8
**Confidence:** 4

**Summary:**

The paper addresses the retrieval component of the retrieval-augmented generation. It argues that existing RAG methods tend to focus exclusively on semantic similarity or relational links, leading to suboptimal performance in complex, multi-hop reasoning tasks. The paper motivates this with a coverage study to show that similarity or relatedness can only return a few correct entity connections, and the return results are half overlapped. The paper proposes SIRERAG, which is designed to optimize information retrieval by indexing data based on similarity and relatedness. The index is based on a similarity and relatedness tree using a a method similar to RAPTOR's.

Experiments with three datasets (MuSiQue, 2WikiMultiHopQA, and HotpotQA) showed improvements in EM and F1 scores compared to previous indexing methods, which include similarity or relatedness. The gain in performance had a slight negative impact on the inference time compared to a similarity-only method due to a larger retrieval pool size. Further analyses showed that the relatedness tree was indeed helpful.

**Strengths:**

1. The paper's main idea is well-motivated and supported by its preliminary experiments (Table 1). I found that its arguments were well-articulated and supported by existing literature, yet the hard evidence provided in Table 1 was a helpful addition.
2. The paper acknowledged an alternative approach in some steps and some results to reject the alternatives (Sections 4.1 and 4.3).
3. The method proposed in the paper consistently improved over three multi-hop reasoning datasets (Tables 4 and 6). In addition, ablation analysis also justified the need for the relatedness tree (Table 5).
4. The paper included an inference time experiment, showing that inference time increased with pool size.

**Weaknesses:**

1. Although not related directly to the paper, the paper should still acknowledge the state-of-the-art multi-hop reasoning method such as [Open-RAG](https://openragmoe.github.io/).
2. Section 4.3, which described an alternative method rather than the main method, did not explain flattened indexing well. Figure 2 showed an "+" sign but was not technical enough to confirm what was being done.
3. A few prompts were not provided, such as the summary prompt (maybe similar to RAPTOR?), the topic extraction prompt (Section 3), and the hierarchy prompt (Section 4.1).
4. Some missing details of the baseline might be crucial to interpret the results (see Questions)

**Questions:**

1. It is unclear whether you re-ran the baseline or obtained the results from the original papers. I could not find the exact numbers reported elsewhere in Table 4.
2. What was the retrieval size of the baselines?
3. Overall, I found that TPRS was not meaningful. Wasn't it simply a result of whether the documents were short or long?

------
After the discussion, the score was raised from 6 to 8.

---

> ### Author Response · Authors · 2024-11-25
> **Response to Reviewer gpao**
>
> We thank the reviewer for the helpful feedback! We plan to submit another draft that incorporates all reviewers’ comments before the discussion phase ends.
>
> **Weakness 1**: Thanks for your suggestion! We notice that the Open-RAG paper was public on arxiv after we made this ICLR submission. We will cite it in our revised version. As shown in Table 6 and the scores below (our responses to reviewer F1mv and ymw3), SiReRAG successfully improves the performance of other non-indexing methods on multihop reasoning. Therefore, we think SiReRAG has the potential of complementing Open-RAG, because the two methods are not in conflict.
>
> |               | MuSiQue EM | MuSiQue F1 | 2Wiki EM | 2Wiki F1 | HotpotQA EM | Hotpot F1 | Average EM | Average F1 |
> |---------------|------------|------------|----------|----------|-------------|-----------|------------|------------|
> | Self-Ask      | 31.20      | 44.35      | 55.00    | 61.99    | 57.10       | 71.11     | 47.77      | 59.15      |
> | SiReRAG + Self-Ask | **36.50** | **49.12** | **57.20** | **65.13** | **59.70**  | **74.07** | **51.13**  | **62.77**  |
>
> **Weakness 2**: Flattened indexing is a simple method that places all tree nodes into a unified retrieval pool. In other words, regardless of a node's origin (e.g., bottom or upper levels, similarity or relatedness trees), it is added to a single list containing all nodes. We will clarify this concept in Section 4.3 of our revised version.
>
> **Weakness 3**: We provide our prompts as below. We will add these prompts in our appendix.
>
> Summary prompt (same as RAPTOR): `Summarize the provided text, including as many key details as needed.`
>
> Topic extraction prompt (Section 3): `Identify the high-level topic of this paragraph as concise as possible.`
>
> Hierarchy prompt (Section 4.1):
> ```Generate content structure for the document.
> ## Instructions
> 1. Group sentences based on their topic and abstractiveness.
> 2. For each sentence group define metadata consisting of topic and the abstractiveness level.
> 3. There are two abstractiveness levels:
>     a. low: sentences describe fine-grained details about the topic
>     b. high: sentences gives an overview of the topic, summarizing fine-grained details.
> 4. Ensure that each group consists of contiguous sentences.
> 5. Respond using the below json format.
> {
>   "group_1":
>   {
>     "sentences": [list of sentence ids],
>     "topic": "topic of the group",
>     "abstractiveness": "high"
>   },
>   "group_2":
>   {
>     "sentences": [list of sentence ids],
>     "topic": "topic of the group",
>     "abstractiveness": "low"
>   },
>   "group_3":
>   {
>     "sentences": [list of sentence ids],
>     "topic": "topic of the group",
>     "abstractiveness": "high"
>   }
> }
> ```
>
> **Weakness 4 and Question 1**: We reran all baselines, which is important to ensure the fair comparison of our experiments. For example, we retrieve top 20 candidates that match the provided query and use GPT-4o to answer queries (Section 5.4) for all methods.
>
> **Question 2**: As mentioned in line 368, we retrieve 20 candidates for all models including all the baselines and SiReRAG.
>
> **Question 3**: Thanks for your comment! We do not claim SiReRAG as the most efficient RAG indexing method (accuracy on multihop QA tasks is our priority in this work). Having a larger retrieval pool than RAPTOR, we aim to study the efficiency of SiReRAG from a different angle than inference time. Inference complexity involves many factors, including length of documents and redundancy of added tree nodes. Therefore, we will design a new metric called Time-Pool Efficiency Ratio (TPER): $\frac{\text{Inference-time A} / \text{Inference-time B}}{\text{Pool-size A} / \text{Pool-size B}}$. This metric computes the growth of total inference time with respect to the growth of the retrieval pool size between two methods. In other words, we aim to ensure that the growth of inference time does not scale proportionally with the increase in the retrieval pool size. A TPER value less than 1 indicates reasonable efficiency of method A compared with method B, whereas a TPER value greater than 1 signifies lower efficiency. Thus, we set SiReRAG as method A and RAPTOR as method B to compare their efficiency.
>
> |  | MuSiQue | 2Wiki  | HotpotQA |
> |---------------------------------------------------------|---------|--------|----------|
> | TPER (using SiReRAG as method A and RAPTOR as method B) | 0.600   | 0.499  | 0.517    |
>
> We see that these TPER values are well below 1, which indicates that SiReRAG is a reasonably efficient method.

---

> > ### Comment · Reviewer_gpao · 2024-11-26
> > **Thank you for your rebuttal**
> >
> > The authors addressed my concerns quite well. Additional data answered most of my questions, so I am more confident in recommending this paper for the conference. I have no further questions.
> >
> > Reviewing the response to Weakness 1 of Reviewer F1mV, I found it quite interesting regarding the generalizability of the proposed technique in other domains where entity and graph might not be easily obtained. I believe from the standpoint of the paper, the authors responded well.

---

> ### Author Response · Authors · 2024-11-26
> **Thanks for your acknowledgment**
>
> Thanks so much for your kind acknowledgment and raising your score!😊 As mentioned, we are working to upload a revised PDF by the end of November 27 to incorporate all reviewers' comments.

---

### Official Review · Reviewer_gA7K · 2024-11-04

**Soundness:** 4
**Presentation:** 4
**Contribution:** 3
**Rating:** 8
**Confidence:** 4

**Summary:**

This paper presents SiReRAG: a new indexing and retrieval technique that takes into account both "semantic similarity" and "entity relatedness" for answering complex multi-hop queries.

The suggested technique is built mostly on top of another technique called RAPTOR (which indexes text chunks on similarity only), but augments it by: first, extracting entity-level facts or propositions s (which are single facts/statements about entities), using LLMs like chatGPT or LLama-instruct, then applying the RAPTOR pipeline to extracted propositions to create relatedness trees akin to the original RAPTOR similarity trees. The final method merges the output from the two trees (one from original RAPTOR built on text chunks, and another from RAPTOR built on extracted and aggregated propositions).

The paper includes comprehensive evaluations and shows significant improvement over baseline methods that use only similarity or relatedness features.

**Strengths:**

Originality: The work presented in SiReRAG is original in the sense it combines ideas from two existing methods/philosophies about indexing, similarity and entity-relatedness into one solution.

Quality: The thought process and reasoning of the paper is mostly intuitive and simple (merge two ideas that have been shown to improve performance), the experimental section supports this reasoning and is somewhat comprehensive with high quality and well-studied datasets.

Clarity: The paper is mostly well-written and easy to follow, with certain exceptions are included in the Weaknesses sections. The claims of the authors about coverage of relatedness or similarity only in table 1 serves as a good benchmark and motivation for the work.

Significance: I consider this method to be an incremental addition on top of RAPTOR, The main metrics reported show significant improvement over the baseline methods that use only one signal.

**Weaknesses:**

- The paper is not very clear regarding how to merge the result from the two trees (similarity and relatedness trees), but I might have missed this.

- The metric "time per retrieval pool size" which is used to quantify the time of the SiReRAG vs other baselines method seems very contrived and not at all relevant, since it seems to say more about the number of candidates (the denominator) than about the time spent answering a query. Which means that the number of candidates generated by this method can be an order of magnitude more than the baseline methods in some cases, which can be prohibitive in many use-cases.

- The main body of the paper has only two examples, without much detail: I recommend adding a full simple example for how the relatedness tree would look for a simple paragraph, as it's more difficult to visualize the entity-related tree as opposed to the similarity/summary between text chunks.

- The suggested method ends up performing badly on the more structured datasets (2Wiki): this indicates that the way it's encoding the relatedness is not representative enough to capture the structure of the graph/triples/facts.

**Questions:**

- The reasoning behind why HippoRAG does significantly better for 2Wiki dataset is not very clear in my opinion: I think what might be happening here is the following: when the RAPTOR text clustering pipeline is applied to the extracted entity-relatedness propositions, it treats the full propositions as text, completely ignoring the underlying structure of the proposition/fact (which is usually of the form Subject/Predicate/Object). That destroys certain information from the triple. My guess is that if we modify the RAPTOR encoder to encoder these features separately (e.g. with SPO markers tokens or with different encoders altogether), then we could see that we're able to encode the triple structure better, and we might be able to recover that difference.

- It would be very informative to share more information about the average number of candidates considered by the method compared to the baseline methods per query. This would show the computational increase compared to the baseline.

---

> ### Author Response · Authors · 2024-11-25
> **Response to Reviewer gA7K (Part 1)**
>
> We thank the reviewer for the helpful feedback! We plan to submit another draft that incorporates all reviewers’ comments before the discussion phase ends.
>
> **Weakness 1**: We mention to flatten nodes for retrieval, and we will clarify this concept in Section 4.3 of our revised version. Flattened indexing is a simple method that places all tree nodes into a unified retrieval pool. In other words, regardless of a node's origin (e.g., bottom or upper levels, similarity or relatedness trees), it is added to a single list containing all nodes.
>
> **Weakness 2**: Thanks for your comment! We do not claim SiReRAG as the most efficient RAG indexing method (accuracy on multihop QA tasks is the priority). Having a larger retrieval pool than RAPTOR, we aim to study the efficiency of SiReRAG from a different angle than inference time. Inference complexity involves many factors, including length of documents and redundancy of added tree nodes. We will design a new metric called Time-Pool Efficiency Ratio (TPER): $\frac{\text{Inference-time A} / \text{Inference-time B}}{\text{Pool-size A} / \text{Pool-size B}}$. This metric computes the growth of total inference time with respect to the growth of the retrieval pool size between two methods. In other words, we aim to ensure that the growth of inference time does not scale proportionally with the increase in the retrieval pool size. When we set SiReRAG as method A and RAPTOR as method B, a TPER value less than 1 indicates reasonable efficiency, whereas a TPER value greater than 1 signifies low efficiency. Thus, we have TPER values to compare SiReRAG and RAPTOR.
>
> |  | MuSiQue | 2Wiki  | HotpotQA |
> |---------------------------------------------------------|---------|--------|----------|
> | TPER (using SiReRAG as method A and RAPTOR as method B) | 0.600   | 0.499  | 0.517    |
>
> We see that these TPER values are well below 1, which indicates that SiReRAG is a reasonably efficient method.
>
> **Weakness 3**:
> Thanks for your suggestion! We will add a full example of the relatedness tree in the appendix of our revised draft. We will briefly describe a tree example here. For the multihop question in Figure 1 (correct answer: Nicholas Bacon), the MuSiQue corpus contains one relevant paragraph stating, 'Francis Bacon was born on 22 January 1561 at York House near the Strand in London, the son of Sir Nicholas Bacon…'. The RAPTOR tree for the entire MuSiQue corpus has two more mentions (both mentions are in summary nodes) of Nicholas Bacon, one of which reads: 'Francis Bacon: Born on 22 January 1561, son of Sir Nicholas Bacon and Anne Cooke Bacon.' The addition of our relatedness tree adds two more mentions of Nicholas Bacon: one is in a proposition aggregate (“Francis Bacon was the son of Sir Nicholas Bacon, Lord Keeper of the Great Seal, and Anne (Cooke) Bacon.\n…**Head I** is a small oil and tempera on hardboard painting by Francis Bacon, completed in 1948.…”), and the other one is a summary node (“The text provides detailed genealogical and biographical information about several individuals from different families and historical periods… Francis Bacon was the son of Sir Nicholas Bacon and Anne (Cooke) Bacon, making William Cecil, 1st Baron Burghley, his uncle...”). Thus, RAPTOR tree has three mentions of Nicholas Bacon, but none of them contains Head I information. SiReRAG has five mentions of Nicholas Bacon, and one of them (the proposition aggregate) contains Head I information. This proposition aggregate groups several propositions together via the entity “Francis Bacon”. Because this node is the only retrieval candidate that fully matches and answers the question, retrieving it would maximize the chance of generating the correct answer. If we use the RAPTOR tree only, we will not have this retrieval candidate. We believe this is an excellent example of how a comprehensive knowledge integration process can enhance the performance of LLMs in multihop reasoning. As a result, we are motivated to model both signals of similarity and relatedness.
>
> **Weakness 4 and Question 1**: Thanks for your great suggestion! Yes, we agree with you that the RAPTOR text clustering pipeline does not have any special mechanism to take care of the underlying structure of proposition aggregates, which is a future direction of our work. Encoding nodes differently provides a promising approach to further enhancing SiReRAG's performance. Although 2Wiki’s entity-centric nature is well-suited for HippoRAG, we would like to highlight that our method yields significantly better performance than HippoRAG on average. As this paper focuses on modeling both similarity and relatedness, tailoring text encoding to capture fine-grained details would be a valuable extension.

---

> ### Author Response · Authors · 2024-11-25
> **Response to Reviewer gA7K (Part 2)**
>
> **Question 2**: As stated in line 368, we retrieve the top 20 candidates for each query in both SiReRAG and the baselines for fair comparisons. The retrieval pool sizes of SiReRAG on MuSiQue, 2Wiki, and HotpotQA are 35070, 19100, and 29934 respectively. The retrieval pool sizes of RAPTOR on MuSiQue, 2Wiki, and HotpotQA are 12371, 6939, and 10031 respectively. SiReRAG's retrieval pool size is slightly less than three times the size of RAPTOR's. Considering the discussion above on TPER, we believe our method is reasonably efficient.

---

> ### Comment · Reviewer_gA7K · 2024-11-26
>
> Thank you for addressing the comments and explaining more. I look forward to reading the modified draft.
>
> After this discussion, I am raising my soundness and presentation scores to 4.

---

> > ### Author Response · Authors · 2024-12-03
> >
> > Thanks a lot for your response and valuable feedback! We uploaded our modified draft a few days ago and will include additional results on the ASQA dataset (sent through our general post) in our final version.

---

### Official Review · Reviewer_ymw3 · 2024-11-11

**Soundness:** 3
**Presentation:** 2
**Contribution:** 2
**Rating:** 6
**Confidence:** 4

**Summary:**

The paper introduces SiReRAG, which proposes to consider both similar and relatedness information when creating retrieval indices that address queries with multi-hop reasoning. The paper gives motivations to demonstrate the bottleneck of solely modelling relatedness or similar information only. For similar information, the paper constructs a similarity tree based on recursive summarization, while for relatedness, SiReRAG extracts propositions and entities from text, and groups them via shared entities to construct a relatedness tree. Experimental results show SiReRAG considerably improves over other baselines like HippoRAG, GraphRAG and RAPTOR when evaluated on various multi-hop QA datasets.

**Strengths:**

1)	The approach shows considerable performance improvements over RAPTOR on a variety of multi-hop QA datasets.

2)	The experimental ablation settings are thorough and show the benefit of different design choices made by the authors for the SiReRAG approach.

**Weaknesses:**

1)	The paper does read like incremental work over RAPTOR, and it is hard to be convinced that the paper has enough novelty for acceptance at ICLR.

2)	Some important baselines such as the closed book approach, i.e. directly getting the final answer from the LLM without any retrieval, or iterative retrieval, such as Self-Ask [1] or DSPy [2] are missing. Also, the authors should include these baselines when doing the inference latency comparison.

3)	The writing needs to be improved in Sections 1 and 3 of the paper since it’s not easy to grasp the main intuitions or motivations of SiReRAG.

[1] Measuring and Narrowing the Compositionality Gap in Language Models

[2] DSPY: COMPILING DECLARATIVE LANGUAGE MODEL CALLS INTO SELF-IMPROVING PIPELINES

**Questions:**

1)	The motivation example in Figure 1 is a bit misleading. The authors suggest that the question “Who is the father of the artist who painted Head I?” has three hops of reasoning, whereas it only requires two hops, i.e. finding the artist in 1st hop and then identifying the author in the 2nd hop. This example can be a bit confusing to the reader on what the overall motivation of the approach is.

2)	It is very surprising to see almost similar performance for models of highly different capabilities such as GPT-3.5-turbo and GPT-4o. The authors should provide more insight/analysis on why this is happening? Also, what is the performance when the LLM used is an open-source model such Llama-3 8B.

3)	It would be interesting if the authors could show performance separately based on the number of reasoning hops present in the question. Also, does the approach show any benefits over RAPTOR for single hop/simple queries?

4)	The authors should also evaluate more recent multi-hop QA datasets, such as FanOutQA or FreshQA. All the datasets considered are pre-2022, raising concerns about leakage into LLM pretraining data.

5)	The authors should also consider adding a few qualitative analysis examples that demonstrate how and “why” (i.e. which part of the method helps) SiReRAG improved over RAPTOR due to incorporating the relatedness.

---

> ### Author Response · Authors · 2024-11-25
> **Response to Reviewer ymw3 (Part 1)**
>
> We thank the reviewer for the helpful feedback! We plan to submit another draft that incorporates all reviewers’ comments before the discussion phase ends. Focusing on multihop reasoning, we are, to the best of our knowledge, the first to verify the importance of incorporating both similarity and relatedness signals and to implement this concept in RAG indexing.
>
> **Weakness 1**:
> To the best of our knowledge, our work is the first indexing method of modeling both similarity and relatedness in RAG setup for a more comprehensive knowledge integration. Our baselines (RAPTOR, HippoRAG, and GraphRAG) consider only a single perspective. In contrast, we find that modeling both perspectives enables a more effective knowledge integration process, as detailed in Section 3 (adopting a clustering strategy based on a single perspective covers a remarkably low percentage of supporting passages). Our quantitative results in Section 6 echoes this motivation.
>
> Although we are motivated, it also requires nontrivial efforts to realize our high-level idea as a RAG indexing method. Firstly, we decide to focus on multihop reasoning as our scope, as knowledge integration or synthesis is vital to this kind of task as discussed in Figure 1. Then, we need to finalize an effective and efficient structure (e.g., tree/knowledge graph) to index corpus data. RAPTOR tree is a straightforward idea, but it deviates from the commonsense understanding of a tree structure [1]. As noted in line 207, while multiple levels of data abstraction can exist within a document, RAPTOR places all text chunks at the bottom level. This approach conflicts with many studies on document discourse trees [2]. Then, we explore the hierarchical structure of an indexing tree. However, we do not see a clear trend of improvement by doing so (Table 2). So, we use RAPTOR as the generalized tree structure for similar and related information. We believe our exploration and observation are valuable for future work.
>
> As stated, we adhere to a rigorous thought process. The effectiveness (Table 4 and our response to Question 3) and wide applicability (Table 6 and our response to Weakness 2) of SiReRAG is a significant contribution, as reasoning capability is crucial for LLMs, and multihop reasoning questions are among the most complex types of queries.
>
>
>
> [1] Zhang, J., Silvescu, A., & Honavar, V. (2002). Ontology-driven induction of decision trees at multiple levels of abstraction. In Abstraction, Reformulation, and Approximation: 5th International Symposium, SARA 2002 Kananaskis, Alberta, Canada August 2–4, 2002 Proceedings 5 (pp. 316-323). Springer Berlin Heidelberg.
>
> [2] Maekawa, A., Hirao, T., Kamigaito, H., & Okumura, M. (2024, March). Can we obtain significant success in RST discourse parsing by using Large Language Models?. In Proceedings of the 18th Conference of the European Chapter of the Association for Computational Linguistics (Volume 1: Long Papers) (pp. 2803-2815).

---

> > ### Author Response · Authors · 2024-11-25
> > **Response to Reviewer ymw3 (Part 2)**
> >
> > **Weakness 2**:
> > Although there are many existing methods that work on multihop reasoning tasks, SiReRAG is about indexing corpus data under RAG setup. In other words, instead of being our baselines, other existing works focus on other dimensions of improving performance on complex reasoning tasks. To the best of our knowledge, we have selected a wide variety of RAG indexing baselines to demonstrate our claims.
> >
> > However, we have run two additional sets of experiments: (1) the closed-book setting, and (2) an iterative retrieval method (Self-Ask) specifically designed for multihop reasoning. Without using a search engine, we prompt GPT-4o in two iterations to implement Self-Ask. In the first iteration, the model is prompted to propose follow-up questions and provide answers to them. In the second iteration, GPT-4o is instructed to answer the final question by incorporating the follow-up thought process. We feed the model with a one-shot example and 10 retrieved candidates that match the final question in both iterations. Similar to Table 6, our experiment on Self-Ask shows that SiReRAG can complement existing methods for optimal performance.
> >
> > |               | MuSiQue EM | MuSiQue F1 | 2Wiki EM | 2Wiki F1 | HotpotQA EM | Hotpot F1 | Average EM | Average F1 |
> > |---------------|------------|------------|----------|----------|-------------|-----------|------------|------------|
> > | Closed Book   | 10.0       | 22.0       | 19.0     | 34.0     | 29.0        | 44.0      | 19.3       | 33.3       |
> > | Self-Ask      | 31.20      | 44.35      | 55.00    | 61.99    | 57.10       | 71.11     | 47.77      | 59.15      |
> > | SiReRAG + Self-Ask | **36.50** | **49.12** | **57.20** | **65.13** | **59.70**  | **74.07** | **51.13**  | **62.77**  |
> >
> > We see that the closed-book setting yields the worst performance. Although these three datasets are pre-2022 and data leakage into LLM pre-training data might happen, LLMs’ parametric knowledge alone does not offer a decent performance on these datasets. Then, we see SiReRAG successfully improves the scores of Self-Ask, demonstrating its wide applicability. By leveraging SiReRAG's retrieval pool, we view our method as an augmentation to other non-indexing methods for multihop reasoning, rather than as a competitor.
> >
> > Efficiency-wise, we also show average time per query (TPQ) below. By having SiReRAG, the TPQ of Self-Ask increases by approximately 1.2 seconds over the three datasets. Since Self-Ask requires two iterations of LLM prompting in our implementation, the increase in TPQ is relatively small compared to the significant performance improvement brought by SiReRAG.
> >
> > |                   | MuSiQue TPQ | 2Wiki TPQ | HotpotQA TPQ | Average TPQ |
> > |-------------------|-------------|-----------|--------------|-------------|
> > | Self-Ask          | 2.72        | 2.21      | 2.29         | 2.41        |
> > | SiReRAG + Self-Ask| **4.53**    | **3.07**  | **3.30**     | **3.63**    |
> >
> > **Weakness 3**: Thanks a lot for your comment! We will improve the writing of those sections in our revised version. To summarize our motivation, we notice that indexing both similar and related information facilitates a more comprehensive knowledge integration process than solely modeling one perspective.
> >
> > **Question 1**: We will fix our writing of our motivation example. We meant to associate chunks with important entities, because we can possibly retrieve at least three different kinds of chunks for the multihop question provided. Therefore, instead of using terms such as 'hop 1 chunk,' we will refer to it as “entity 1 chunk”. We aim to use this example to showcase the benefit of considering both similarity and relatedness for a more comprehensive knowledge integration.
> >
> > **Question 2**: We discussed this phenomenon in lines 417 to 420. As mentioned in Section 5.4, we use GPT-4o to handle QA for all methods, since we only focus on indexing. When it comes to building the indexing tree or knowledge graph, different LLMs might be used. Thus, in line 367, we mention that “we use either GPT-3.5-Turbo or GPT-4o as the choice of LLM if any methods involve LLM calls”. Here we would like to clarify that these “LLM calls” refer to calls used by indexing only. Both RAPTOR and SiReRAG show similar performance on two different LLMs, because the choice of LLMs only determines summarization output for these two methods. As this is not a challenging summarization task (e.g., summarizing a small cluster of nodes that are similar and/or related), the choice of LLMs for indexing does not play an important role for these two methods, which allows researchers to choose a cheaper LLM. However, when we switch to a different model to handle QA, it is expected that the performance would vary significantly, but our scope is RAG indexing.

---

> > > ### Author Response · Authors · 2024-11-25
> > > **Response to Reviewer ymw3 (Part 3)**
> > >
> > > **Question 3**:
> > > Thanks for your suggestion! We run a set of experiments on single-hop questions. Specifically, we use the MuSiQue dataset and collect all the decomposed questions of multihop queries. We filter out some decomposed questions if they are still multihop or are based on another question. As a result, we end up with 502 single-hop questions from MuSiQue. Performance on them is shown below.
> > >
> > > |           | EM    | F1    |
> > > |-----------|-------|-------|
> > > | Raptor    | 37.10 | 40.50 |
> > > | SiReRAG   | **37.20** | **41.49** |
> > >
> > > SiReRAG still outperforms RAPTOR, but the lead narrows compared to the scores in Table 4. Since all queries in MuSiQue involve at least two hops, we observe that an increased number of reasoning hops positively impacts SiReRAG's performance. This is because single-hop questions may not require comprehensive knowledge synthesis, as they only involve retrieving the relevant chunks for the single hop. However, with more hops, we not only need to retrieve relevant chunks but also synthesize them comprehensively.
> > >
> > > **Question 4**:
> > > To showcase the generality of SiReRAG on more multihop QA datasets, we have tried MultiHopRAG recently. We show the performance of SiReRAG and RAPTOR on this newer dataset. We filter all unanswerable questions and randomly select 350 “comparison” queries and 350 “inference” queries, which forms a pool of 700 queries in total.
> > >
> > > |         | MultiHopRAG EM | MultiHopRAG F1 |
> > > |---------|----------------|----------------|
> > > | RAPTOR  | 59.90          | 60.35          |
> > > | SiReRAG | **61.00**      | **61.69**      |
> > >
> > > SiReRAG still delivers better performance on MultiHopRAG, which echoes our main experiment. Our comprehensive selection of datasets demonstrates the generality of SiReRAG on multihop reasoning tasks.
> > >
> > > As mentioned above (in question 2), we use GPT-4o to handle QA for all methods, making our experiment a fair comparison, even when considering data leakage. Moreover, we discussed in weakness 2 that the closed-book setting yields poor performance, which further mitigates the leakage concern. Same as HippoRAG, we reported performance on MuSiQue, 2Wiki, and HotpotQA in our original draft.
> > >
> > > **Question 5**:
> > > Yes, we will add a qualitative example of the relatedness tree in the appendix of our revised draft. We will briefly describe a tree example here. For the multihop question in Figure 1 (correct answer: Nicholas Bacon), the MuSiQue corpus contains one relevant paragraph stating, 'Francis Bacon was born on 22 January 1561 at York House near the Strand in London, the son of Sir Nicholas Bacon…'. The RAPTOR tree for the entire MuSiQue corpus has two more mentions (both mentions are in summary nodes) of Nicholas Bacon, one of which reads: 'Francis Bacon: Born on 22 January 1561, son of Sir Nicholas Bacon and Anne Cooke Bacon.' The addition of our relatedness tree adds two more mentions of Nicholas Bacon: one is in a proposition aggregate (“Francis Bacon was the son of Sir Nicholas Bacon, Lord Keeper of the Great Seal, and Anne (Cooke) Bacon.\n…**Head I** is a small oil and tempera on hardboard painting by Francis Bacon, completed in 1948.…”), and the other one is a summary node (“The text provides detailed genealogical and biographical information about several individuals from different families and historical periods… Francis Bacon was the son of Sir Nicholas Bacon and Anne (Cooke) Bacon, making William Cecil, 1st Baron Burghley, his uncle...”). Thus, RAPTOR tree has three mentions of Nicholas Bacon, but none of them contains Head I information. SiReRAG has five mentions of Nicholas Bacon, and one of them (the proposition aggregate) contains Head I information. This proposition aggregate groups several propositions together via the entity “Francis Bacon”. Because this node is the only retrieval candidate that fully matches and answers the question, retrieving it would maximize the chance of generating the correct answer. If we use the RAPTOR tree only, we will not have this retrieval candidate. We believe this is an excellent example of how a comprehensive knowledge integration process can enhance the performance of LLMs in multihop reasoning. As a result, we are motivated to model both signals of similarity and relatedness.
> > >
> > > Our analysis in Section 6.2 also presents the benefits of SiReRAG components quantitatively through ablation study.

---

> > > > ### Comment · Reviewer_ymw3 · 2024-11-28
> > > > **Response to Author Rebuttal**
> > > >
> > > > Thank you for addressing my concerns. I have appropriately raised my final score.

---

> > > > > ### Author Response · Authors · 2024-12-03
> > > > >
> > > > > Thanks a lot for your response! If there are still concerns about the generality and contribution of SiReRAG, we provided additional results on a different dataset called ASQA in our general post. Thanks again for your valuable feedback!

---

### Author Response · Authors · 2024-11-28
**Our draft has been revised to incorporate all comments.**

We sincerely thank all the reviewers for their valuable feedback. We have incorporated their comments in the main content or appendix of our modified draft. We just updated our PDF. Below are the major modifications:

1. We added Self-Ask as another non-indexing method to showcase the wide applicability of SiReRAG.

2. We added our experiment on MultiHopRAG dataset and single-hop questions of MuSiQue dataset to showcase the generality of SiReRAG.

3. We added our experiment of “entity clustering” and allowing cross-tree interaction to demonstrate a principled design of our method.

4. We replaced TPRS with TPER to better study the efficiency of SiReRAG from a different angle than inference time.

5. We added the closed-book setting to test the capability of LLMs’ parametric knowledge on multihop questions.

6. We provided more details of SiReRAG such as the prompts we used and a tree example.

7. We updated the performance of GraphRAG and HippoRAG to show our good-faith effort of maximizing baselines’ scores.

8. We updated our writing (mainly in Section 1 and 3) to emphasize our contribution and improve overall clarity.

---

### Author Response · Authors · 2024-12-03
**Additional Results on the Generality and Contribution of SiReRAG**

To further showcase the generality of our relatedness tree and SiReRAG, we compare the performance of RAPTOR and SiReRAG on a dataset different from the multihop reasoning datasets. We run both methods on ASQA (Answer Summaries for Questions which are Ambiguous) [1], and this dataset contains factoid questions that are ambiguous. Thus, the primary difference between the selected multihop QA datasets and ASQA is that ASQA requires LLMs to reason across multiple perspectives (e.g., disambiguated questions) of an ambiguous question and organize their generation into a coherent and detailed answer. Indexing all knowledge pieces of the development set of ASQA via RAPTOR or SiReRAG, we show performance scores below.

|          | String Exact Match | Disambig-F1 | Disambiguation-Rouge |
|----------|--------------------|-------------|-----------------------|
| RAPTOR   | 51.47             | 40.30       | 42.72                |
| SiReRAG  | **52.81**         | **41.82**   | **43.47**            |

These scores are reported based on all 948 ambiguous questions of the development set. Specifically, string exact match (STR-EM) and Disambig-F1 dissect RAG answers to ambiguous questions into multiple perspectives and measure their accuracy with respect to each disambiguated question. Disambiguation-Rouge serves as an overall statistic that incorporates both ROUGE (with respect to the references) and accuracy scores. We see that SiReRAG delivers a consistent improvement over RAPTOR, which again demonstrates the benefit of adopting SiReRAG on complex reasoning queries.

In conclusion, SiReRAG's generality across various complex reasoning tasks (e.g., multihop QA, single-hop QA, and ambiguous questions) is well demonstrated, with the most significant improvement observed in multihop QA. Therefore, our paper mainly focuses on multihop reasoning, and we will include this empirical analysis on ASQA in our final version to show the significant contribution of SiReRAG on complex reasoning tasks.



[1] Stelmakh, I., Luan, Y., Dhingra, B., & Chang, M. W. (2022, December). ASQA: Factoid Questions Meet Long-Form Answers. In Proceedings of the 2022 Conference on Empirical Methods in Natural Language Processing (pp. 8273-8288).

---

### Meta-Review · Area_Chair_qCnc · 2024-12-19

**Metareview:**

This paper presents SiReRAG, a novel RAG indexing approach that explicitly considers both semantic similarity (of text embeddings) and entity relatedness (measured by keyword/entity overlap) for improved performance on complex multihop reasoning tasks. The method extends RAPTOR's similarity-based tree structure and complements it with a relatedness tree built from entity-specific propositions. Both trees are flattened into a unified retrieval pool for downstream use.

The reviewers agreed that the paper is well-motivated (reviewers ymw3 and gpao), presents clear empirical evidence for the need to model both similarity and relatedness signals (reviewers gpao and gA7K), and shows consistent improvements over strong baselines across multiple datasets (all reviewers). The experimental validation is thorough, with comprehensive ablation studies and analysis (reviewers gA7K and gpao). The only remaining weakness is arguably its potential lack of generalization beyond entity-centric tasks.

Based on the strong experimental results, thorough evaluation, and convincing responses to reviewer concerns, the paper is recommended for acceptance.

**Additional Comments On Reviewer Discussion:**

Some initial concerns were raised about the novelty beyond RAPTOR (reviewer ymw3), the lack of certain baselines like Self-Ask (reviewer ymw3), and the generalizability beyond entity-centric tasks (reviewer F1mV). During the discussion period, the authors thoroughly addressed these concerns through additional experiments and clarifications. The authors demonstrated the method's effectiveness by adding experimental results with Self-Ask and closed-book settings, providing additional validation on MultiHopRAG and ASQA datasets, and including detailed ablation studies that demonstrated the benefits of their design choices.

Overall, the paper achieves strong improvements over state-of-the-art methods. While reviewer F1mV remained somewhat concerned about the method's specialization to entity-centric tasks, the other reviewers were convinced by the authors' thorough responses and additional experiments demonstrating broader applicability.

---

### Decision · Program_Chairs · 2025-01-22

Accept (Poster)